



# On the influence of erect shrubs on the irradiance profile in snow

Maria Belke-Brea[1,2,3], Florent Domine[1,2,4,*], Ghislain Picard[5], Mathieu Barrere[1,2,3,6], and Laurent Arnaud[5]

[1]Takuvik Joint International Laboratory, Université Laval (Canada) and CNRS-INSU (France), Québec City, QC, G1V 0A6, Canada.
5   [2]Centre d'Études Nordiques, Université Laval, Québec City, QC, Canada.
[3]Department of Geography, Université Laval, Québec City, QC, Canada.
[4]Department of Chemistry, Université Laval, Québec City, QC, Canada.
[5]Univ. Grenoble Alpes, CNRS, IRD, Grenoble INP, IGE, Grenoble, France.
[6]Météo-France – CNRS, CNRM UMR 3589, CEN, Grenoble, France.

10   *Correspondence to*: Florent Domine (florent.domine@gmail.com)

**Abstract.** The warming-induced expansion of shrubs in the Arctic is transforming snowpacks into a mixture of snow, impurities and buried branches. Because snow is a translucent medium into which light penetrates up to tens of centimeters, buried branches may alter the snowpack radiation budget with important consequences for the snow thermal regime and microstructure. To characterize the influence of buried branches on radiative transfer in snow, irradiance profiles were 15   measured in snowpacks with and without shrubs near Umiujaq in the Canadian Low Arctic (56.5° N, 76.5° W) in November and December 2015. Using the irradiance profiles measured in shrub-free snowpacks in combination with a Monte Carlo radiative transfer model revealed that the dominant impurity type was black carbon (BC) in variable concentrations up to 185 ng g$^{-1}$. This allowed the separation of the radiative effects of impurities and buried branches. Irradiance profiles measured in snowpacks with shrubs showed that the impact of buried branches was generally weak, except for layers where branches 20   were also visible in snowpit photographs, suggesting that branches influence snow locally (i.e. a few centimeters around branches). The local-effect hypothesis was further supported by observations of localized melting and depth hoar pockets that formed in the vicinity of branches. Buried branches therefore affect snowpack properties, with possible impacts on Arctic flora and fauna and on the thermal regime of permafrost. Lastly, the unexpectedly high BC concentrations in snow are likely caused by nearby open-air waste burning, suggesting that cleaner waste management plans are required for northern 25   community and ecosystem protection.

## 1 Introduction

Due to Arctic warming, erect shrubs are expanding into the tundra biome, replacing low-growing vegetation like grasses, lichen and mosses (Tape et al., 2006; Myers-Smith et al., 2011; Ropars and Boudreau, 2012; Lemay et al., 2018). The vegetation change is transforming natural snowpacks, which originally consisted of snow with impurities, to a mix of snow, 30   impurities and branches (Pomeroy et al., 2006; Loranty and Goetz, 2012). This has a large influence on the snow radiation budget, because branches are much more light-absorbing than snow in the visible range (Juszak et al., 2014; Belke-Brea et





al., 2019). Numerous experimental and model-based studies have investigated the albedo-reducing effect of branches that protrude above the snow surface (e.g. Sturm et al., 2005; Pomeroy et al., 2006; Liston et al., 2002; Loranty et al., 2011; Ménard et al., 2014, Belke-Brea et al., 2020). However, little attention has been given to the potential effects of branches that

are buried in the snowpack.

Snow is a translucent medium into which light can penetrate 20 to 40 cm deep, depending on the wavelength and snow physical properties (France et al., 2011; Tuzet et al., 2019). Light penetration and transmittance are important parameters influencing photochemical processes (Grannas et al., 2007; Domine et al., 2008; France et al., 2011) and the thermal regime of the snowpack (Flanner and Zender, 2005; Picard et al., 2012). In turn, the thermal regime controls snow melt rates in

spring and during warm spells in autumn, which is of crucial importance for many bio-geophysical processes in the tundra ecosystem and for Arctic climate (Walker et al., 1993). For example, snow melt timing impacts hydrological processes (Pomeroy et al., 2006), permafrost thawing (Romanovsky et al., 2010; Johansson et al., 2013), energy and mass exchanges between the surface and the atmosphere (Groendahl et al., 2007), hibernation behavior of Arctic fauna (Berteaux et al., 2017; Domine et al., 2018) as well as the growing season length of Arctic flora (Cooper et al., 2011; Semenchuk et al., 2016).

Moreover, the depth of light penetration and the amount of transmitted light both impact the microstructure of the snowpack: they influence the formation of melt-freeze grains and the degree of temperature gradient metamorphism (Aoki et al., 2000; Domine et al., 2007). Because the insulating properties of a snowpack depend on its microstructure, light distribution in snow could ultimately affect the permafrost  thermal regime and its thawing rate due to climate change (Pelletier et al., 2018). These complex processes highlight the importance of studying snow-light interactions in the snowpack and

understanding how buried branches may alter these processes.

In natural snowpacks, light propagation is strongly influenced by light-absorbing particles (LAP) but also by buried branches. Studying the radiative forcing of LAP in snow, identifying typical LAP types, and quantifying LAP concentrations on a global scale has been an active field of study over the last decades (e.g. Warren and Wiscombe, 1980; Hansen and Nazarenko, 2004; Doherty et al., 2010; Skiles et al., 2018, Tuzet et al., 2019). It is now known that LAPs increase light

absorption in the UV and visible spectrum (350–750 nm), where the absorption by ice is extremely weak, but that their effect is negligible in the near-infrared spectrum (>1000 nm) where ice itself is sufficiently absorptive (Picard et al., 2016; Warren, 2019). Each type of LAP has a specific wavelength-dependent absorption efficiency (Fig. 1), which creates a characteristic shape in plots of spectral absorption measured in snow (Bond et al., 1999; Grenfell et al., 2011; Dal Farra et al., 2018). Due to this spectral signature, optical measurements can be used to not only separate different types of LAP but also measure

their respective concentrations in snowpacks. The most absorbing impurity commonly found in snow is black carbon (BC), but significant concentrations of mineral dust are also found in windy and mountainous regions or close to deserts (e.g. Ramanathan et al., 2001; Painter et al., 2007; Moosmüller et al., 2009; Dang et al., 2017). BC concentrations across the Arctic vary between 4 ng g$^{-1}$ in Greenland and 60 ng g$^{-1}$ in Arctic Russia and Scandinavia (Doherty et al., 2010). The principal source of BC over wide areas in the Arctic is the anthropogenic emissions due to the incomplete combustion of

biomass and fossil fuels (Diehl et al., 2012; Bond et al., 2004). The polluted air is transported to the Arctic by atmospheric



circulation where it deposits on the snowpack in varying concentrations (from 5 to 50 ng g$^{-1}$; Rahn and McCaffrey, 1980; Cess, 1983). Additionally, there is an increasing number of local sources of BC in the Arctic, like ship emissions or the open-air burning of waste that further increase BC air concentrations (Abbatt et al., 2019). Besides LAP, buried branches also have an effect on light transmission and absorption because branches are highly light-absorbing in the visible spectrum (Fig. 1).

However, this effect has not yet been studied, mainly because erect vegetation was mostly absent in high latitudes and high elevation environments, which coincidentally, is where snow is a dominant factor (Stevens and Fox, 1991; Holtmeier and Broll, 2007). However, shrubs are now expanding northwards due to Arctic warming, and the effect of buried branches on the snow radiation budget in the Arctic tundra may gain in importance.

Irradiance in snow and the effect of LAPs are generally computed numerically with radiative transfer models (Warren and

Wiscombe, 1980; Hansen and Nazarenko, 2004; Aoki et al., 2011; Tuzet et al., 2017). Today, it is possible to calculate radiative transfer through snow as a function of snow physical properties (i.e. snow density and specific surface area (*SSA*)), using the analytical equations established by Kokhanovsky and Zege (2004). The radiative effect of impurities is calculated for pre-established impurity concentrations from the optical properties that are associated to the different impurity types (i.e. the impurity-specific mass absorption efficiency, MAE). Models have calculated that at concentrations in the 5–50 ng g$^{-1}$

range BC typically reduces albedo from 0 % to 4 %. Albedo reductions of 1–4 % can cause positive radiative forcing of 4–16 W m$^{-2}$ during an average Arctic spring and early summer day, and hence are climatically significant (Warren, 2019). Albedo measured over mixed surfaces with snow and protruding branches showed considerably higher albedo reductions of up to 30 % in the broadband (300–2800 nm) and up to 55 % at 500 nm where the contrast between snow and vegetation is most extreme (Sturm et al., 2005, Belke-Brea et al., 2019). These reductions were mostly associated to light absorption by

protruding branches and it remains to be tested whether the impact of buried branches in the Arctic tundra could be of the same magnitude as the LAPs mixed in the snowpack. From a radiative transfer modeling point of view, branches and LAP are very different objects. The latter is homogeneously mixed with snow so that its absorption can be averaged and combined with that of the ice, and the classical solution of the radiative transfer equation for homogeneous media applies without any change. In contrast, branches are macroscopic embedded absorbers that affect the path of light, a situation that has no simple

analytical solution. To design a model that accurately represents buried branches and allows calculating their specific radiative impact, it is first necessary to acquire basic knowledge about how snow, light and buried branch interact.

This study aims to bring the first insights, to our knowledge, on how buried branches influence light propagation in snowpacks. We present a qualitative analysis where we use a combination of *in situ* measurements and radiative transfer simulations. The latter were computed with the radiative transfer model SnowMCML (Picard et al., 2016) for snowpacks

with known snow physical properties and estimated impurity type and concentrations. *In situ* data were acquired during a field campaign in Umiujaq, Northern Quebec (56.5° N, 76.5° W), in autumn 2015 and consisted of (i) vertical spectral irradiance profiles (350–900 nm) measured in snowpacks with and without shrubs, and (ii) vertical profiles of snow density and *SSA* measured in snowpits. Impurity concentrations and types were estimated by applying two methods which allow retrieving impurity information from optical measurements taken in shrub-free snowpacks. The first method consisted in





evaluating the fit between measured and simulated irradiance profiles, where simulated profiles were computed with a range of different impurity types and concentrations. For the second method, extinction coefficients were determined from measured irradiance profiles by using linear regression and the spectral shape of the extinction coefficients was then analysed in order to retrieve information on impurities (Tuzet et al., 2019). The effect of buried branches was investigated by comparing irradiance profiles measured in shrub snowpacks with SnowMCML simulations that include LAPs and irradiance

profiles in shrub-free snowpacks.

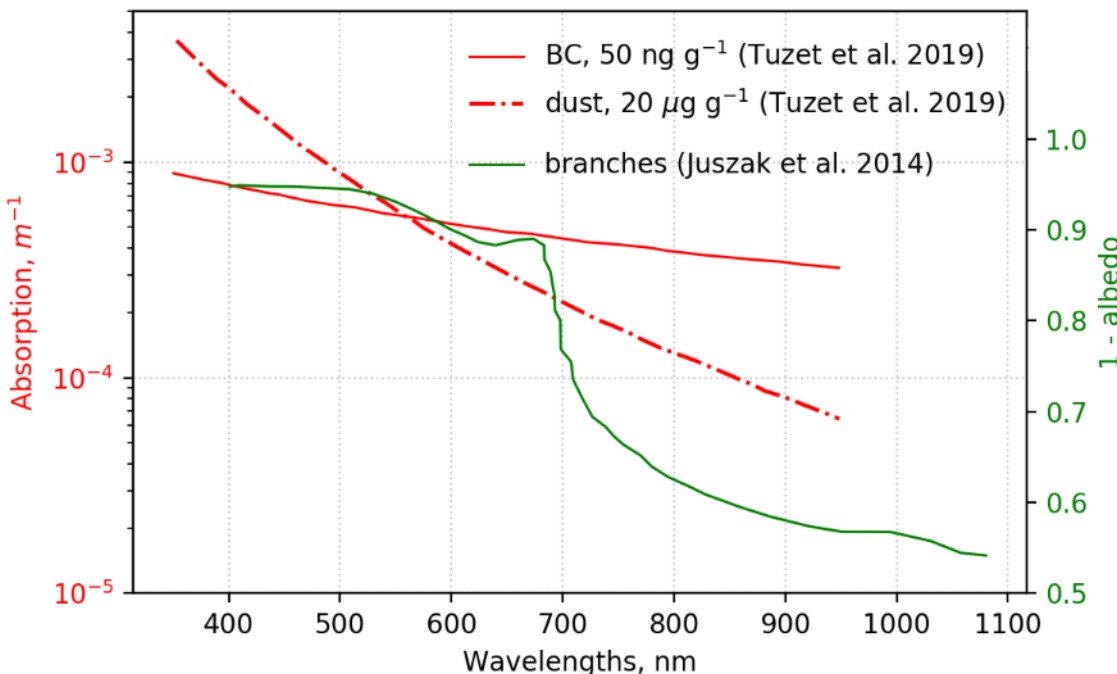

**Figure 1: Comparison of the spectral absorption of BC (red), dust (red dotted) and branches (green). Absorption of branches is illustrated by co-albedo measurements (Juszak et al. 2014). Branch absorption is strongly wavelength dependent and decreases sharply for wavelengths >680 nm.**

## 110    2 Study site and data acquisition

### 2.1 Study site

Our study site is located near the village of Umiujaq on the coast of Hudson Bay in Nunavik, Northern Quebec (56°33′07″ N, 76°32′57″ E, Fig. 2). Measurements were taken in Tasiapik Valley, ~4 km from Umiujaq village. The study sites happen to be also ~2 km from a waste disposal site, where waste was occasionally burned in open-air conditions. The measuring

sites were situated on a wind-exposed plateau. The plateau is covered with lichen and shrubs, but spruces (mainly *Picea mariana*) are also found in wind-sheltered depressions. Over the last three decades, Nunavik has experienced the strongest





greening trend in North America (Ju and Masek, 2016). This is due to shrubs expanding in the tundra biome which are replacing lichen patches of mostly *Cladonia spp.* (Ropars and Boudreau, 2012; Provencher-Nolet et al., 2014; Gagnon et al., 2019). In the Umiujaq region, the main shrub species are birches (*Betula glandulosa*), willows (mostly *Salix glauca* and *S.*

*planifolia*) and alders (*Alnus viridis* subsp. c*rispa*). An automatic weather station has been recording climatic data in the Tasiapik Valley since 1997 (Fig. 2). From 1997 to 2018, the mean annual air temperature has been -3°C (CEN, 2018). In the region of Umiujaq, strong winds and snowstorms are frequent, and wind speeds can reach up to 100 km h$^{-1}$ (Barrere et al., 2018). The predominant wind direction is from the bay (west and north-west) (Paradis et al., 2016) and our measuring sites are thus mostly downwind from the village and the waste burning site (Fig. 2). After data analysis it seemed very likely that

fumes from the waste burning in open-air could have reached our measuring sites and probably affected the acquired data when wind speeds were high enough.

**Figure 2: Map of the study area in the Tasiapik Valley near the village Umiujaq. The blue rectangle marks the area where irradiance profiles were measured in shrub-free snowpacks. The red dots mark where irradiance profiles were measured in snowpacks with shrubs. A white cross marks the position of the Automatic Weather Station (AWS) and a red star marks the site**

**where waste was burned. Map source: Natural Resources Canada (http://atlas.gc.ca/toporama/en/index.html).**





Data were acquired during a field campaign from 29 October to 6 December 2015. During that period, snow and irradiance measurements were taken in four snowpacks with shrubs and three shrub-free snowpacks. Measurements in snowpacks with shrubs were conducted on 3, 9, 14 and 23 November. Snow height at these sites varied between 43 and 63 cm and shrub height varied between 60 and 100 cm (Table 1). Measurements in shrub-free snowpacks were conducted on 8, 22 and 28

November, and snow height at these sites varied between 18 and 30 cm (Table 1). We aimed to conduct measurements for shrub and shrub-free snowpacks at weekly intervals, but harsh measuring conditions and frequent blizzards often prevented us from maintaining this regular measuring interval.

**Table 1. Average snow height and shrub height in Umiujaq for the three shrub-free snowpacks and the four snowpacks with shrubs.**

| Site | Date | Shrub height, cm | Snow height, cm | Protruding branches, cm |
|---|---|---|---|---|
| Snow only | 08 Nov 2015 | - | 18 | - |
| | 22 Nov 2015 | - | 23 | - |
| | 28 Nov 2015 | - | 30 | - |
| Shrubs | 03 Nov 2015 | 60 | 43 | 17 |
| | 09 Nov 2015 | 100 | 58 | 42 |
| | 14 Nov 2015 | 80 | 65 | 15 |
| | 23 Nov 2015 | 60 | 50 | 10 |

**2.2 Data acquisition**

**2.2.1 Spectral irradiance profiles**

Vertical irradiance in the snowpack was measured with the SOLar EXtinction in Snow profiler (SOLEXS). The instrument was developed and tested by Libois et al. (2014) and Picard et al. (2016), where a full description and schematic illustrations

can be found. Basically, the SOLEXS instrument consists of an optical fiber cable which is inserted into a metallic rod painted in white (color: RAL 9003). The rod (10 mm diameter) is vertically inserted in the snowpack into a hole of the same diameter which was punched by a metal rod prior to the measurement. Throughout the continuous manual descent and subsequent rise of the rod in the hole, its position is registered with a depth sensor with a resolution of 1 mm. The optical cable is connected to an Ocean Optics MayaPro spectrometer with a spectral range of 300 to 1100 nm and a resolution of 3

nm. Here, we use measurements from 350 to 900 nm only because the signal-to-noise ratio is too low outside this range. Spectral radiation is recorded every 5 mm while the rod is continuously moving down and up the hole. The maximum





acquisition depth is ~40 cm. Below 40 cm the signal-to-noise ratio becomes too low because of the reduced light intensity, and the shadow of the operator cannot be neglected past this point (Libois et al., 2014), as detailed in Picard et al. (2016). The acquisition of one irradiance profile took around 2 minutes once the instrument was deployed. A photosensor was placed

at the snow surface to monitor the incident radiation changes during the acquisition. If changes in incident radiation exceeded 3 %, the measurement was discarded. Measurements were conducted during any lighting conditions, i.e. overcast, partially overcast and sunny.

SOLEXS is accompanied by a post-processing library (Picard et al., 2016). This library automatically deploys the following processing to the recorded profiles: 1) subtraction of the dark current, 2) a depth correction using the small difference of

timestamps between the depth and spectrum acquisitions, and 3) normalization by the photosensor current to correct for the small fluctuation of irradiance during the complete acquisition.

### 2.2.2 Snowpit data

After each acquisition of a SOLEXS profile, we dug a snowpit at the same spot. In the snowpit, the snow stratigraphy was recorded and photographed, vertical profiles of snow density and snow specific surface area (*SSA*) were measured, and, in

snowpacks with shrubs, the presence of branches was noted. Snow density profiles have a resolution of 3 cm and were measured with a 100 cm³ box cutter (Domine et al., 2016). *SSA* is the surface area of the snow-air interface per mass unit and is inversely related to the optical grain diameter of snow (Warren, 1982; Domine et al., 2007). *SSA* was acquired with the DUFISS instrument detailed in Gallet et al. (2009). Briefly, DUFISSS measures the infrared reflectance of snow samples at 1310 nm by using an integrating sphere. *SSA* is then calculated from that reflectance with a simple algorithm (Gallet et al.,

2009). *SSA* profiles were measured with a resolution of 1 to 3 cm.

Knowing snow density and *SSA* allows calculating the light absorption efficiency of snow by using radiative transfer theory (Kokhanovsky and Zege, 2004; Picard et al., 2016). For these calculations, density and *SSA* need to be available with the same depth resolution. Where this was not the case in our data set, we performed linear interpolation between measured density data points, in order to synchronize the *SSA* and density profiles.

### 175  3 Methodology

#### 3.1 Overview of methods

SOLEXS records the irradiance intensity at different depths (*I(z)*) and thus shows how much light is transmitted through the snowpack. The intensity of transmitted light decreases with depth either because radiation gets absorbed or because it is scattered which provokes a change in the light path direction. The processes of scattering and absorption together are called

extinction. In a pure snowpack, extinction is mainly due to scattering. In contrast, impurities in the snowpack as well as buried branches cause light to become extinct mainly through absorption. Hence, when referring to light-impurity or light-



branch interactions, for all practical purposes extinction and absorption can be used synonymously. In this study we are interested in comparing the extinction of light with depth in snowpacks with and without shrubs. This extinction is visualized as log-irradiance profiles ($I_{log}(z, \lambda)$):

$$I_{\log}(z, \lambda) = \log(I_0(\lambda) / I(z, \lambda)) \quad , \tag{1}$$

where $I_0$ is the incoming radiation at the surface, $z$ is snow depth, $\lambda$ is wavelength, and ($I(z, \lambda)$) are the measured SOLEXS profiles. Hence, to obtain $I_{log}(z, \lambda)$ from the measured data, SOLEXS profiles were normalized with $I_0(\lambda)$ and then presented in log scale. Here, the surface irradiance values for normalization are obtained at a depth of 3 cm ($z = -3$), because the presence of direct light may influence measurements at shallower depths.

Snowpacks are heterogeneous media made up of several kinds of light-extinctive materials – i.e. snow, light-absorbing particles (LAP) and, for snowpacks with shrubs, buried branches. During SOLEXS acquisitions, the measuring rod inserted into the snowpack also contributes to light extinction (Picard et al., 2016). If the interaction between the different light-extinctive materials is negligible, the log-irradiance profile in the medium $I_{log}(z, \lambda)$ is the sum of the material-specific terms. In snowpacks with shrubs this is calculated as:

$$I_{\log}(z, \lambda) = E_{snow}(z, \lambda) + E_{rod}(z, \lambda) + E_{LAP}(z, \lambda) + E_{shrub}(z, \lambda) \quad , \tag{2}$$

where $E_{snow}$, $E_{LAP}$, $E_{shrub}$ and $E_{rod}$, represent the material-specific extinction of snow, impurities, shrubs and the measuring rod, respectively. In order to evaluate the extinction due to buried branches $E_{shrub}$ from measured $I_{log}$ profiles, $E_{snow}$, $E_{rod}$ and $E_{LAP}$, need to be calculated or estimated.

The approach presented here is a simplification, and more physically based approaches can be imagined where the influence
of branches would be calculated with sophisticated 3-D radiative transfer models. Such an approach would require to perform complex simulations and to precisely characterize the optical and physical properties of our medium (i.e. the snowpack with branches and impurities). However, at this stage, very little is known about the influence of branches on radiative transfer in snow. We therefore gave precedence to this simpler and more straightforward approach to obtain first insights into the buried branches-snow-light interactions.

To determine the amount of light absorption by branches, we applied three successive steps. (i) First, light extinction by snow and the measuring rod ($E_{snow} + E_{rod}$) was calculated with a radiative transfer model as a function of *in situ* measured snow physical properties (Sect. 3.2). (ii) Next, the light absorption of impurities ($E_{LAP}$) was estimated using two complementary methods that allow retrieving impurity information from the irradiance profiles measured in shrub-free snowpacks (Sect. 3.3). (iii) Finally, based on the information acquired on $E_{snow}$, $E_{rod}$, and $E_{LAP}$ in the two previous steps, we
determined the influence of buried branches in the irradiance profiles measured in snowpacks with shrubs using Eq. (2).



## 3.2 Calculation of light extinction by snow and the measuring rod

The 3-D radiative transfer model SnowMCML was used to compute the combined light extinction of snow and the measuring rod ($E_{snow}$ + $E_{rod}$). SnowMCML was developed by Picard et al. (2016) and is based on the model "Monte Carlo modeling of light transport in multi-layered tissues" (MCML) from Wang et al. (1995). Specifically, Picard et al. (2016)

adapted the model to compute the signal recorded at the tip of a rod inserted in a multi-layered snowpack. The snow physical properties of each snow layer are supposed to be known, as well as the absorption of the rod. A detailed description of the model is given in Picard et al. (2016). Briefly, the model traces $N$ light rays through a multi-layered snowpack with known physical properties. At each calculation step, light absorption and scattering is determined and the associated decrease in intensity for each light ray is calculated. To optimize calculation time, the model uses the inverse principle in optics,

launching rays from the collectors tip and tracing them back to the source at the surface instead of launching rays at the surface. Using this inverse mode allows to calculate only the path of those rays that hit the collector and which are thus relevant to compute the signal recorded at the tip of the rod. The size and optical properties of the rod need to be known to implement its effect in the simulations. Following the indications from Picard et al. (2016), the rod was modeled as a cylinder with a 10 mm diameter and a length corresponding to the insertion depth of the rod. The albedo of the rod ($\omega_{rod}$) was

set to 0.9 based on the reflectance measurements of the paint conducted by Picard et al. (2016), and it was assumed that the rod had Lambertian scattering characteristics, i.e. rays hitting the rod are scattered in a random direction. In the simulations, the rays hitting the rod were absorbed with a probability 1-$\omega_{rod}$.

In addition to the size and optical properties of the rod, input data to SnowMCML were the physical properties of snow measured in the snowpits. The model outputs were theoretical transmittance profiles. These profiles show light transmittance

for a snowpack without LAPs and branches, but with the same physical properties ($SSA$ and density) as the snowpacks investigated. The simulated $I(z, \lambda)$ profiles were normalized and converted to log scale to obtain the log-irradiance profiles ($I_{log}(z, \lambda)$) from Eq. (1). These profiles were then compared with the log-irradiance profiles acquired in the field.

At transition zones, the performance of the model was found be limited. These zones include the snow-atmosphere transition in the uppermost layer, the transition between two stratigraphic layers inside the snowpack, or the transition from

snow to the underlying soil layer at the bottom of the snowpack. Discrepancies at the snow-atmosphere transition are probably caused by the rod entering the snowpack and causing an optical disturbance. Moreover, close to the surface and down to -7 cm, direct light can potentially penetrate the snowpack and come in through holes around the measuring rod as detailed by Picard et al. 2016.  Since the presence of direct light is known to perturb the measured irradiance profiles (Picard et al., 2016, Tuzet et al., 2019), we discarded the first 7 cm in measured and simulated log-irradiance profiles. At

stratigraphic transitions inside the snowpack, a mismatch between the model and the measured log-irradiance can be due to uncertainties in the snow physical properties that are used as input to the model. These arise because snow physical properties often change gradually or are heterogeneous in stratigraphic transition zones. These fine-scale changes are not captured by our measuring profiles with a resolution of 1 to 3 cm, leading in turn to inaccuracies in the simulations compared





to processes in the natural snowpack. Moreover, rod-light interactions that are calculated in the model also depend on snow
physical properties, which can further amplify the discrepancies between the simulated and measured irradiance profiles.

An additional particularity at stratigraphic transition zones is the occasional occurrence of positive irradiance gradients
(Picard et al., 2016). These are caused by different interactions between the rod and radiation in each snow layer near the
layer transition. These interactions are complex and are explained in detail by Picard et al. (2016). Intuitively, when the rod
reaches a lower layer, the magnitude of its artefact is determined by interactions between the rod and the layer above and in
some cases this can result in an increase in the measured signal, even though the radiative transfer in a 1D layered-media
excludes positive irradiance gradients. Although SnowMCML accounts for the rod artefact at transition zones and can
calculate associated positive gradients, their occurrence in simulated and measured profiles may not concur if there are
uncertainties in the snow physical properties that are used as input to the model. For this reason, we excluded transition
zones, i.e. the top and bottom of each layer, from the interpretation of SnowMCML simulations.

### 3.3 Estimation of absorption by LAPs

Determining the specific absorption of LAPs ($E_{LAP}$) with radiative transfer models requires that concentrations of LAP be
given, and that the optical properties for a given impurity type be known. Unfortunately, there are no data on LAPs for the
snowpack near Umiujaq. Therefore, we assumed that LAPs are either mineral dust coming from a local source (hereafter
called dust) or black carbon (BC). Dust can be transported from the cliffs and the barren rock surfaces at the top of cuestas to
the valley during windy autumn storms which are typical for this region (Barrere et al., 2018, Paradis et al., 2016). BC is
typically introduced to Arctic snow through long-range transport from fossil fuel combustion in the south (McConnell et al.,
2007; Doherty et al, 2010). Based on field observations it seems likely that BC was also produced by snowmobile traffic in
the valley and, perhaps more importantly, by the waste burning occurring ~2 km upwind from our study site (Fig. 2). Based
on these assumptions, we employed two methods to estimate the relative concentration of BC and mineral dust.

The first method applies a regression approach to *in situ* data while the second uses SnowMCML to simulate radiative
transfer in a snowpack with impurities. Both methods are similar in that they determine impurity type and concentration. The
advantage of the SnowMCML method is that the model considers the influence of the measuring rod, but the disadvantage is
that it assumes homogeneous impurity concentrations for the entire snowpack. In contrast, the regression method neglects
the impact of the measuring rod but allows determining LAP concentrations for different layers individually. We applied the
two complementary methods to validate results from each other. Finally, both methods were verified against log-irradiance
measured in shrub-free snowpacks, where LAPs were the only unknown light-absorbing material. The two methods are
detailed in the following sections.

### 3.3.1 Regression analysis of experimental profiles

In the first approach, information on LAP concentrations is derived from the irradiance profiles $I(z, \lambda)$ measured with
SOLEXS by analyzing the rate of decrease of light intensity with snow depth following Tuzet et al. (2019). In snow layers





with optically homogeneous conditions, light intensity decreases exponentially (Beer–Lambert Law). After a logarithmic transformation this exponential decrease becomes linear and the rate of decrease for a specific snowlayer is obtained from the slope of a linear regression (Fig. 3a). In the literature, this rate of decrease is commonly referred to as the Asymptotic Flux Extinction Coefficient (e.g. Libois et al., 2013), but for the sake of simplicity we will refer to it as the extinction

coefficient, $k_e$. The rate by which light decreases in the snowpack is wavelength-dependent and $k_e$ is thus usually shown as a spectral curve termed $k_e(\lambda)$ (Fig. 3b). $k_e$ also depends on the physical properties of the snowpack (SSA and density $\rho$), and on the type and concentrations of LAPs mixed in the snowpack. For example, the $k_e(\lambda)$ curve for a dirty snowpack would display higher $k_e$ values than a clean snowpack because the former is a much more absorbing medium and thus absorbs light at a greater rate. Consequently, each snowpack layer has a specific spectral curve $k_e(\lambda)$ which is a function of the physical

properties of the layer as well as the type and concentration of the LAPs mixed in the snowpack. This relation is mathematically expressed as (Libois et al., 2013, Tuzet et al., 2019):

$$k_e(\lambda) = \sqrt{\frac{3(1-g)}{2} \rho^2 SSA \left( \frac{B\gamma_{ice}(\lambda)}{\rho_{ice}} + MAE(\lambda)c \right)} \quad , \tag{3}$$

where $\gamma_{ice}$ is the ice absorption index which was set to the most recent estimate from Picard et al. (2016). $B$ is the ice absorption enhancement factor and $g$ the scattering asymmetry factor, which were set to default values of 1.6 and 0.85,

respectively (Libois et al., 2014). $\rho_{ice}$ is the density of ice (917 kg m$^{-3}$), and finally, $c$ is the LAP concentration in kg kg$^{-1}$ and MAE is the mass absorption efficiency (m$^2$ kg$^{-1}$) describing the optical property of a given LAP type (Caponi et al., 2017). For this study, the impurity type and the associated MAE values were either set to dust or BC. To determine LAP concentrations in the Umiujaq snowpack, the $k_e(\lambda)$ curve deduced from SOLEXS measurements (Fig. 3) ($k_{e\_meas}(\lambda)$) was fitted to the $k_e(\lambda)$ curve calculated with Eq. (3) ($k_{e\_calc}(\lambda)$). To fit both curves, the LAP concentration ($c$ in Eq. (3)) was estimated

using Python's scipy.optimize.least_squares function which minimizes the mean square error between $k_{e\_calc}(\lambda)$ and $k_{e\_meas}(\lambda)$. The final best fit between the $k_{e\_calc}(\lambda)$ and $k_{e\_meas}(\lambda)$ curves in the spectral range considered (350–900 nm) was evaluated with the coefficient of determination ($R^2$) and the error was given by the root mean square error (RMSE). Determination of $k_{e\_meas}(\lambda)$ curves was restricted to layers where the optical properties in the snowpack were homogeneous for at least 3 consecutive centimeters, and the recorded SOLEXS signal was visually linear, because deducing a slope via linear regression

is only possible under these conditions. These layers are hereafter called zones of interest (ZOI). According to Tuzet et al. (2019), ZOIs have to be at least 3 cm thick and lie at a snow depth >7 cm to accurately determine $k_{e\_meas}(\lambda)$. These restrictions are necessary to avoid biases from the SOLEXS measuring rod at shallow depths and around transition zones (discussed in Sect. 3.1; Picard et al., 2016). $k_{e\_meas}(\lambda)$ curves were smoothed using a first order Butterworth filter with the scipy.signal.butter function in Python (cutoff frequency set to 0.05; Fig. 3b).

The spectrum used to fit $k_{e\_meas}(\lambda)$ and $k_{e\_calc}(\lambda)$ ranged from 400 to 450 nm. In this range absorption by ice is lowest, and impurities have the strongest impact on absorption profiles. Constraining the fit to a specific range instead of using the entire




spectrum (350–900 nm) allows testing the hypothesis that BC or dust are the principal impurity types. A good fit between 400–450 nm should also return a good fit at wavelengths >450 nm if the spectral absorption of the absorbers were chosen correctly.


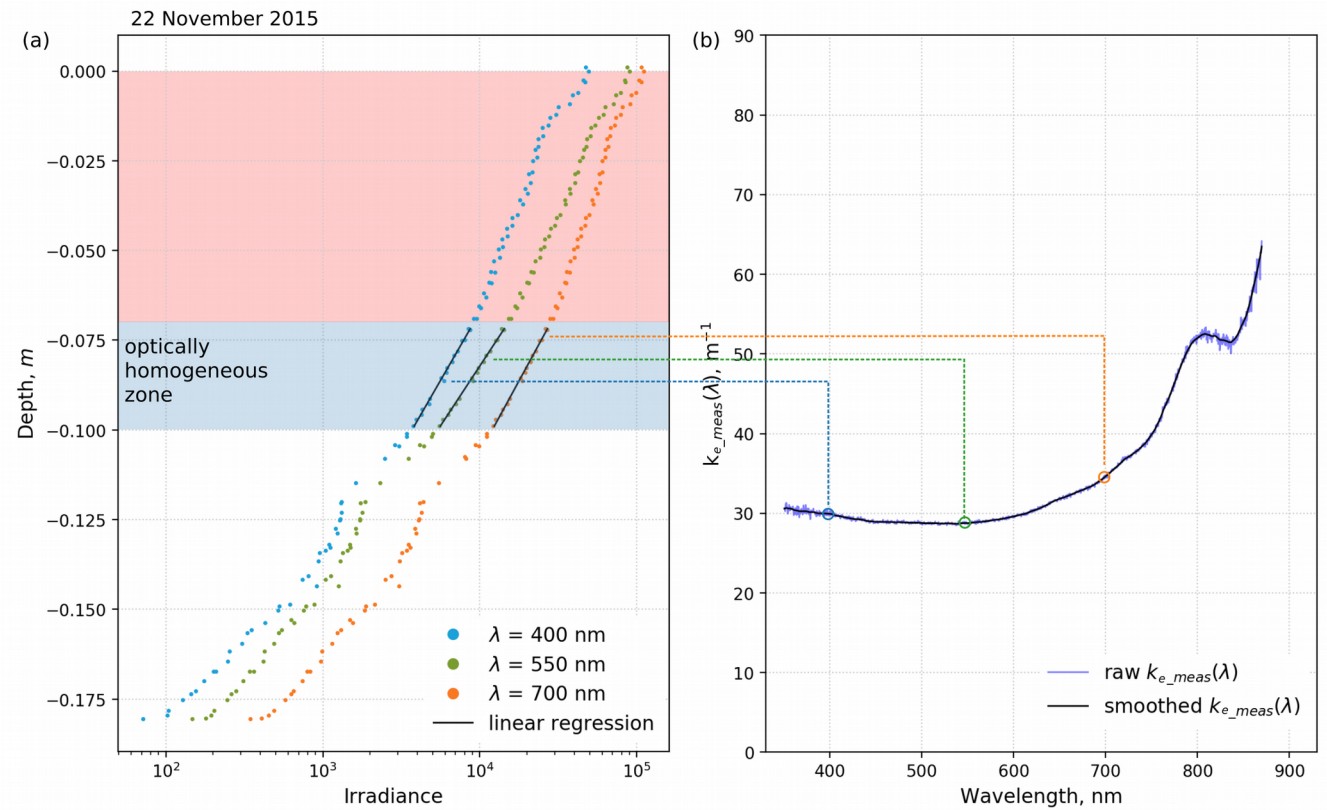

**Figure 3: Overview of how the extinction coefficient ($k_{e\_meas}(\lambda)$) is determined from optically homogeneous layers in irradiance profiles. (a) Irradiance as a function of depth for selected wavelengths. The blue shaded area highlights an optically homogeneous zone where the recorded signal is linear on logarithmic scale. The red shaded area was discarded due to potential influence of direct light. $k_{e\_meas}(\lambda)$ is the slope of the linear regression of irradiance vs. depth in the optically homogeneous zone (black lines). (b)**
**$k_{e\_meas}(\lambda)$ determined for each wavelength in the measured spectrum (350–900 nm) before (blue curve) and after smoothing (black curve). The figure layout was adopted from Tuzet et al. (2019) and modified using data measured near Umiujaq on 22 November 2015.**

### 3.3.2 SnowMCML simulation

SnowMCML allows to simulate the effect of light-absorbing impurities for snowpacks with given LAP concentrations and
MAE values for a given LAP type. For dust, the MAE was taken from Caponi et al., (2017) choosing Algerian dust type, with a grain diameter of 10 μm. Algerian dust was chosen because its optical properties (in particular its absorption Ångstrom exponent) are similar to that of the typical dust reported for snow in the Canadian sub-Arctic (2.5 for Algerian dust vs. 2.2 for sub-Arctic impurities) (Doherty et al., 2010). The relatively large grain diameter of 10 μm (vs. 2 μm) was chosen





because we assumed the dust source to be local. For BC, we followed the approach of Tuzet et al. (2019) and determined the

MAE from the study of Bond and Bergstrom (2006) and Hadley and Kirchstetter (2012). SnowMCML simulations were then computed with a variety of BC and dust concentrations. Note that each simulation corresponds to one LAP concentration as SnowMCML simulated radiative transfer assuming homogeneous LAP concentrations in the entire snowpack. BC or dust concentrations were determined by fitting the simulations with known LAP concentrations to the measured log-irradiance profiles with unknown concentrations. The snow-atmosphere transition zone (0 to -7 cm) and the stratigraphic transition

zones were excluded from the fit as explained above. From the remaining non-transition layers, LAP type and concentrations were deduced from simulations that most accurately represented the radiative effect in the snowpack in Umiujaq. These best-fitting simulations were determined from a visual comparison of the simulated and measured profiles in shrub-free snowpacks.

## 4 Results and discussion

### 4.1 Impurities in snow without shrubs

LAP type and concentrations were determined from log-irradiance profiles measured in shrub-free snowpacks on 8, 22 and 28 November. The two methods, i.e. the $k_e$ analysis and the SnowMCML method, are complementary but they should ideally yield similar results for the deduced LAP type and concentration for the Umiujaq snowpack.

### 4.1.1 LAP type

To test the validity of our initial assumption that LAP was either dust or BC, we compared the fit between $k_{e\_meas}(\lambda)$ and $k_{e\_calc}(\lambda)$ in four zones (ZOI1 to ZOI4) where optical properties were homogeneous and allowed determining an extinction coefficient. Two of these four zones were in snowpits measured on 8 and 22 November, and the other two were in the snowpit of 28 November. We found that setting the LAP type to BC in Eq. (3) constantly returned a very good fit between the estimated and measured $k_e$ curves in all four zones. For example, in ZOI1 which was measured on 8 November, the

achieved fit had a $R^2$ value of 0.98 and a RMSE=1.67 (Fig. 4a) when LAP was set to BC. The fit was good for all wavelengths in the spectrum considered (350–900 nm), suggesting that the spectral absorption signature of BC is well suited to reproduce the extinction coefficients observed in the field data. In contrast, setting the LAP type to dust in Eq. (3) (Fig. 4b) resulted in visibly poor fits between $k_{e\_meas}(\lambda)$ and $k_{e\_calc}(\lambda)$ curves ($R^2$=0.38; RMSE=9.13). The fit was poor for the entire spectrum, but the direction and magnitude of the mismatch was wavelength-dependent, suggesting that the spectral

absorption signature of dust was ill-suited to reproduce the extinction coefficients observed in the field data. Finally, using both BC and dust in Eq. (3) returned results which were essentially the same as for the simulations with BC only, because dust concentrations were estimated to virtually 0, reinforcing the conclusion that LAPs in our study site is mostly composed of BC. These results were similar for the other ZOIs in the shrub-free snowpacks (ZOI2–4), except for ZOI3 where the





BC+dust option returned a fit almost as good as BC only, but still with low dust concentration estimations. Results for all

four ZOIs are listed in Table 2. We conclude from the $k_e$ analysis that BC is the only significant absorber in snow without

shrubs, and the absorption due to dust is negligible.

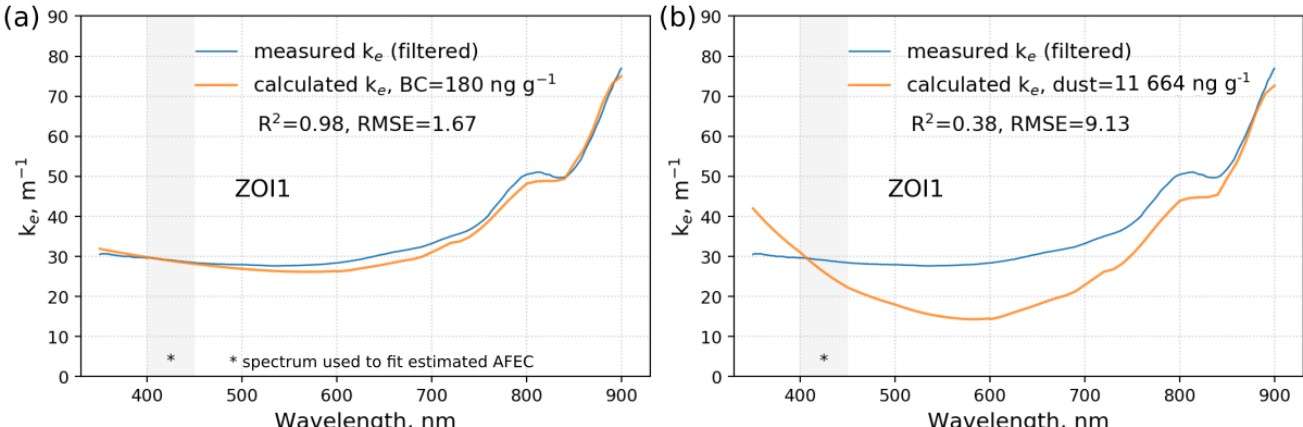

**Figure 4: Example for measured and calculated absorption coefficient $k_e$ for a snowpack without shrubs. Measured $k_e$ was determined from SOLEXS measurements taken on 8 Nov (ZOI1). Calculated $k_e$ was computed with either (a) black carbon (BC)**
**or (b) mineral dust as impurity type in the snowpack. The concentration of dust or BC was determined with an iterative approach, where calculated $k_e$ was fitted to the measured $k_e$. This example shows how assuming BC as impurity type returns significantly better fits.**

The results of the SnowMCML simulations concur with the results of the $k_e$ analysis. Using BC as LAP returned better and

wavelength-independent agreements between the simulated profiles and measured log-irradiance profiles. An example is

shown in Fig. A1 (Appendix 1), demonstrating the fit between measured log-irradiance and SnowMCML profiles with either

BC or dust at 400 and 500 nm. Therefore, from now on we will assume that BC is the dominant impurity type for the

remainder of this study. This result is reasonable because BC has often been found to be the main impurity type in Arctic

snow (e.g. Doherty et al., 2010; Wang et al., 2013). Moreover, the open-air waste burning near our study area was probably

an important additional BC source (Fig. 2). It might be that some trace amounts of dust, coming from the cuestas

surrounding the Tasiapik valley, were also present in the snow, but their impact was too weak to be detected from our optical

measurements. In order to identify all the different LAP constituents, it would be necessary to conduct a detailed chemical

analysis of the snowpack but this was beyond the focus of this work.






**Table 2. Fit between measured and calculated extinction coefficient curves ($k_e(\lambda)$) for measurements in shrub-free snowpacks.**
**Calculated $k_e(\lambda)$ was computed either with black carbon (BC), BC and mineral dust, or mineral dust only. The fit between**
**measured and calculated $k_e(\lambda)$ was analysed with the coefficient of determination ($R^2$), the error is indicated with the root mean**
**square error (RMSE).**

| | | | BC only | | | BC+dust | | | dust only | | |
|---|---|---|---|---|---|---|---|---|---|---|---|
| ZOI | Snow depth [m] | date | ng g$^{-1}$ | $R^2$ | RMSE | ng g$^{-1}$ | $R^2$ | RMSE | ng g$^{-1}$ | $R^2$ | RMSE |
| ZOI1 | -0.7 – -0.1 | 8 Nov. | 180 | 0.98 | 1.67 | BC: 180 Dust: $1.1 * 10^{-25}$ | 0.98 | 1.67 | 11 664 | 0.38 | 9.13 |
| ZOI2 | -0.7 – -0.1 | 22 Nov. | 185 | 0.96 | 2.59 | BC: 184 Dust: $1.2 * 10^{-26}$ | 0.96 | 2.59 | 11 915 | 0.31 | 10.17 |
| ZOI3 | -0.7 – -0.1 | 28 Nov. | 21 | 0.98 | 1.96 | BC: 20 Dust: 42 | 0.98 | 1.98 | 1 359 | 0.96 | 2.84 |
| ZOI4 | -0.14 – -0.17 | 28 Nov. | 7 | 0.91 | 2.02 | BC: 7 Dust: $1.4 * 10^{-18}$ | 0.91 | 2.02 | 450 | 0.93 | 1.85 |

### 4.1.2 LAP concentrations

BC concentrations were found to vary considerably among the shrub-free snowpacks. BC concentrations derived from the $k_e$
analysis varied from 7 to 185 ng g$^{-1}$ (Table 2). On 8 and 22 November, BC concentrations were high with 180 ng g$^{-1}$ and 185
ng g$^{-1}$, respectively. On 28 November the snowpack was comparatively cleaner with only 7 and 21 ng g$^{-1}$ BC in ZOI3 and
ZOI4, respectively.

BC concentrations from SnowMCML simulations were determined from the ZOI layers also used in the $k_e$ analysis (ZOI1–
ZOI4, highlighted in blue in Fig. 5), plus one additional layer (ZOI2_b, Fig.5), while the transition zones (T1–T3 in Fig. 5)
where excluded from our analysis. The additional layer ZOI2_b was only used for the SnowMCML analysis because its
signal-to-noise ratio at longer wavelengths was too weak to establish a spectral $k_{e\_meas}(\lambda)$ curve. Note that in layers ZOI1 and
ZOI2, simulations showed the same extinction gradient as the measured data but with an offset. Consequently, simulated and
measured profiles were parallel to each other in ZOI1 and ZOI2 instead of being superposed. The reason for the offset was
probably that the amount of light transmitted to ZOI1 and ZOI2 from the transition zone was inaccurately calculated by the
model. In non-transition zones and after correcting the offset in ZOI1 and ZOI2, BC concentrations derived by fitting
SnowMCML simulations to measured log-irradiance profiles concurred with the results from the $k_e$ analysis. On 8 and 22
November, SnowMCML simulations with 185 ng g$^{-1}$ fitted best with the observed absorption gradients, while on 28
November the snowpack was cleaner and the best fit was achieved with a simulation without BC.









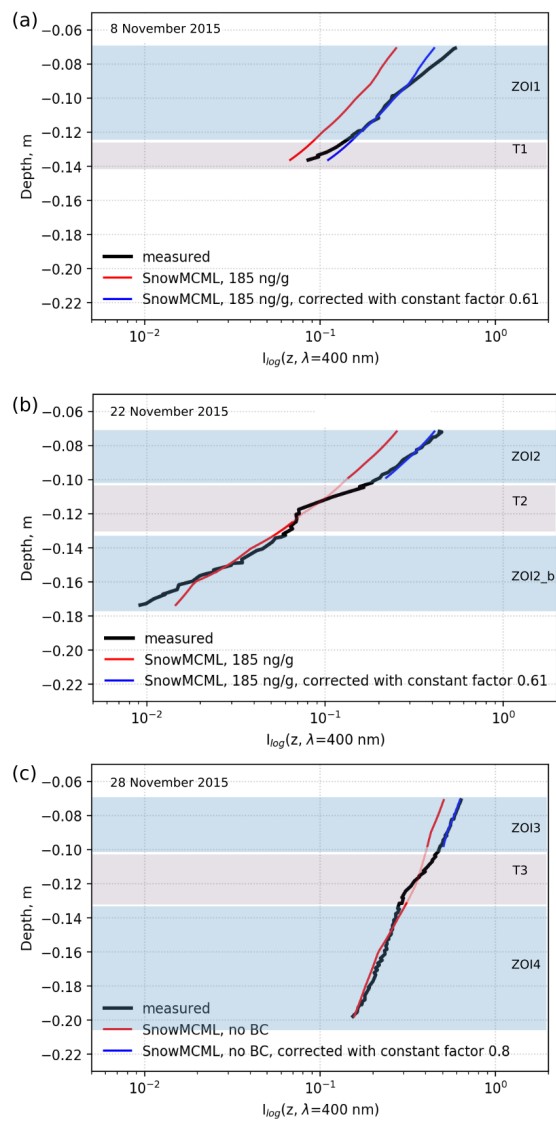

**Figure 5: Measured log-irradiance profiles (black curves) and SnowMCML simulations (red and blue curves) for snowpacks without shrubs at 400 nm. Simulated profiles were computed assuming black carbon (BC) as impurity type. Log-irradiance**
**profiles were measured on (a) 8 Nov., (b) 22 Nov. and (c) 28 Nov. Gray shaded areas highlight transition zones, where simulated and measured profiles were not expected to fit. Blue shaded areas highlight non-transition zones were the fit between simulated and measured profiles allowed the determination of impurity concentrations.**

## 4.2 Insights into the radiative effect of buried branches

Determining the effect of buried branches from the acquired log-irradiance profiles proved a complex task because high BC
concentrations had a strong impact on absorption, potentially masking the effect of branches. Furthermore, we could not deduce a constant impurity concentration representative of the Umiujaq snowpack in general. Consequently, in Eq. (2), $E_{LAP}(z)$ and $E_{shrub}(z)$ both remain unknown variables in the log-irradiance profiles measured in snowpacks with shrubs.



Nevertheless, interesting insights on how buried branches might influence light propagation were gained by (i) comparing SnowMCML simulations with the measured log-irradiance profiles and (ii) studying the spectral shape of $k_{e\_meas}(\lambda)$ and

$k_{e\_calc}(\lambda)$ for different layers in snowpacks with branches.

From the comparison of measured log-irradiance profiles with SnowMCML simulations at 400 nm, we found that snowpacks in shrubby areas consisted of two types of optically distinct layers (Fig. 6). Characteristics of the first layer type were that the measured profiles fitted well with the SnowMCML simulations (we called these layers IMP1 through IMP4 in Fig. 6), although the simulations only considered the extinction of light by snow, the measuring rod and BC, but not by

shrubs. Moreover, photographs of the snowpits showed no or very few branches in these layers. The best examples for this layer type are IMP1 and IMP2, where the measured log-irradiance fitted very well with simulated SnowMCML profiles at BC=100 ng g$^{-1}$ (Fig. 6a and b). The measured log-irradiance profiles in IMP1 and IMP2 decrease linearly, indicating a constant extinction coefficient. These layers thus seem to have homogeneous optical properties and to be free of optical disturbances like branches. In IMP3 and IMP4 the measured log-irradiance profile was less regular, showing numerous small

disturbances in the extinction coefficient. Nevertheless, the general trend in IMP3 fitted well to simulations with BC concentrations of 50 ng g$^{-1}$ and IMP4 to simulations with 200 ng g$^{-1}$. Therefore, it is reasonable to suggest that light absorption in the IMP layers was mostly dominated by BC concentrations and that the influence of branches was negligible.

For the second layer type (BRAN1 through BRAN4, Fig. 6), we found several lines of evidence that branches influenced light absorption in snow. First, unlike in the IMP layers, the log-irradiance profiles did not fit the SnowMCML simulations

well. Secondly, the log-irradiance profiles were very irregular in comparison with IMP1 or IMP2, showing a highly variable extinction coefficient. Finally, comparing the BRAN layers to the snowpit photographs revealed striking correspondences between these layers and the presence of branches. In Fig. 6a, a branch appeared at 22 cm depth, where the simulated and measured profiles start to diverge (BRAN1). Note that between snow depths 34 and 37 cm in BRAN1 the simulated profile shows a positive irradiance gradient which is not visible in the measured signal. As explained in Sect. 3.2 positive gradients

can happen at transition zones and are an artefact caused by the rod. The discrepancy between the measured and modeled profile arises most likely due to uncertainties in the measured snow physical properties input to the model. In Fig. 6b, two small twigs appeared between snow depths 16 to 24 cm, which coincided with a part of the log-irradiance profile that poorly fits the simulations (BRAN2). In this layer the measured profile shows a positive gradient but not the simulated profile. This discrepancy may again be caused by uncertainties in the snow physical properties, but it is more

likely that here it is the result of the optical disturbance caused by branches. Branches absorb light locally thus reducing the irradiance signal, but once the sensor exits the shadow of the branch light may hit it from the side resulting in an increase in measured irradiance and thus an enhancement of the positive gradient. In Fig. 6c, the snowpit had generally more branches than the snowpits in Fig. 6a and 6b. Branches became particularly abundant between 10 and 40 cm depth, as also documented in our field notes. This was also where the simulation started to diverge more significantly from the measured

profile (BRAN3). A high variability of the extinction coefficient, which was already observed in IMP4, was also visible in the irregular log-irradiance profile in BRAN3. Finally, in Fig. 6d, branches were abundant in the entire snowpit and the

 

measured profile could not be properly fitted to any of the simulations (BRAN4). BRAN 4 also showed a highly variable extinction coefficient similar to the log-irradiance profile in Fig. 6c.

**Figure 6: Log-irradiance and SnowMCML simulations at 400 nm for measurements taken on (a) 9 Nov., (b) 3 Nov., (c) 23 Nov., and (d) 14 Nov. in snowpacks with shrubs. Yellow shaded areas highlight layers where measured log-irradiance profiles and SnowMCML simulations fitted well. Green shaded areas highlight layers where log-irradiance and SnowMCML simulations fit less well and branches were visible in the snowpit photographs.**

To further confirm the finding that shrubby snowpacks consist of impurity-dominated and branch-influenced layers, the spectral information from the $k_e$ analysis was exploited. We determined the $k_{e\_meas}(\lambda)$ and $k_{e\_calc}(\lambda)$ curves for IMP1, IMP2 and IMP3 as well as for BRAN1 and BRAN4. With this $k_e$ analysis we aimed to test whether light extinction in IMP layers was indeed dominated by BC concentrations. Furthermore, we aimed to verify that any influence of branches was visible in BRAN1 and BRAN4. Note that $k_e$ curves were not determined for BRAN2, because $k_e$ values were negative due to the observed positive extinction gradient, or for BRAN3 because the signal-to-noise ratio was too low. The results of the $k_e$ analysis are shown in Fig. 7b. For comparison, we also show $k_e$ curves for the ZOIs 1-3 in shrub-free snowpacks (Fig. 7a). For IMP2, $k_{e\_meas}(\lambda)$ and $k_{e\_calc}(\lambda)$ fitted very well ($R^2$=0.97) in the 350–830 nm spectral range and returned BC concentrations

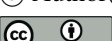



of 92 ng g$^{-1}$ which concurs with the results from the SnowMCML simulations (BC=100 ng g$^{-1}$). We conclude that for IMP2, absorption properties of BC were indeed well suited to reproduce the observed spectral extinction and that the assumption that BC is the dominant absorber is likely. For IMP1 the fit was good for wavelengths <680 nm and estimated BC concentrations (95 ng g$^{-1}$) concurred with SnowMCML results (BC=100 ng g$^{-1}$). However, for wavelengths > 680 nm the two

curves start to diverge and the $k_{e\_meas}(\lambda)$ curve showed a significant drop at wavelengths >780 nm which did not appear in the theoretical $k_{e\_calc}(\lambda)$ curve. For IMP3, BRAN1 and BRAN4 we also observe that the $k_{e\_meas}(\lambda)$ and $k_{e\_calc}(\lambda)$ curves diverge because values in the $k_{e\_meas}(\lambda)$ curves drop at longer wavelengths. For IMP3, BRAN1 and BRAN4 the fit between $k_{e\_meas}(\lambda)$ and $k_{e\_calc}(\lambda)$ was also generally lower with $R^2$ of -4.65 (IMP3), -9.32 (BRAN1) and 0.62 (BRAN4). Note that obtaining negative $R^2$ values is possible because we constrained the fit of $k_{e\_meas}(\lambda)$ and $k_{e\_calc}(\lambda)$ to the range 400–450 nm while the

evaluation was performed for a much larger range. In such a constrained setting, calculated values can fit the observed values less well than a horizontal line (= the null hypothesis) which results in $R^2$ values below 0. The interpretation of negative $R^2$ is that the calculated values fit the observations very poorly in at least part of the spectrum (Motulsky and Christopoulos, 2003).

We interpret the observed drop in the $k_{e\_meas}(\lambda)$ curves at >680 nm as a strong indicator of the influence of branches (Fig. 7).

Reflectivity measurements for Arctic shrub branches showed that branches are highly absorbing at 400–500 nm, but that reflectivity increases slightly at 500 nm and then even more sharply at 680 nm (Juszak et al. 2014) (Fig. 1). We conclude that the optical properties of branches are well suited to explain the observed drop in extinction at 500–900 nm in the measured $k_e$ curves (Fig. 7). In contrast, $k_{e\_calc}(\lambda)$ curves were calculated assuming that all extinction other than by snow or the rod was due to BC. In this case, the $k_{e\_calc}(\lambda)$ curves overestimate extinction in the spectrum >500 nm because in this range BC is

more absorbing than branches (Fig. 1). It is likely that in IMP1 and IMP3 branches had an influence on the irradiance profile although the measured log-irradiance fitted well with the SnowMCML simulations and no branches were detected in the photographs. In BRAN1 to BRAN4 the effect of branches seemed to be stronger, as suggested by the multiple indicators for branch influence (for example irregular profiles or the mismatch between measured log-irradiance and SnowMCML simulations). In contrast, almost no influence of shrubs could be detected in IMP2 despite the layer being located in a

snowpack with shrubs. This leads us to conclude that the optical effect of a buried branch must be highly localized and that its impact strongly weakens as a function of distance from the branch. The log-irradiance profiles here were measured at different distances to branches, but the exact distances are unknown to us, which is why the influence of branches varied in the different IMP and BRAN layers. This shows that quantifying the impact of branches would require to know the position of branches in the snowpack with precision.


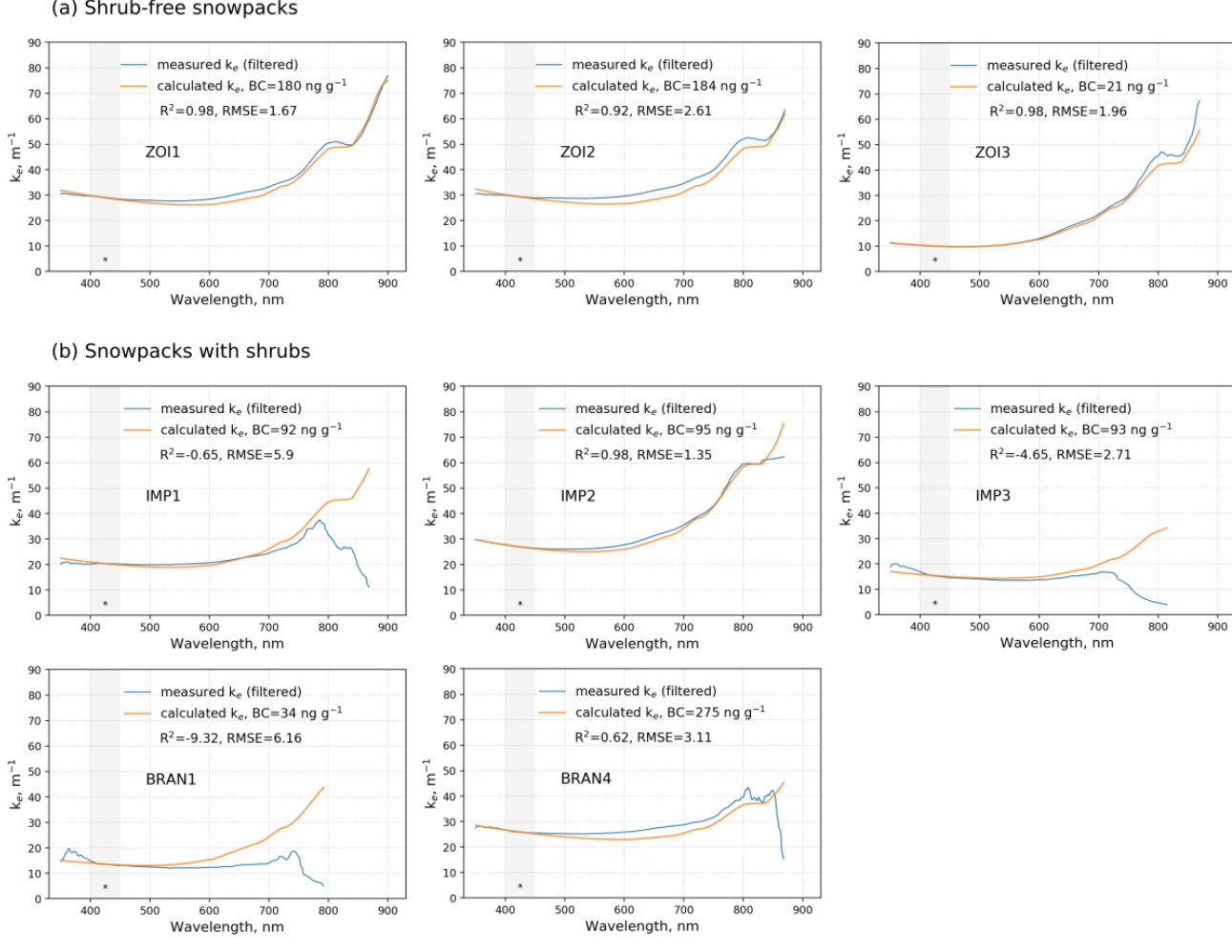

**Figure 7: Measured and calculated $k_e$ for (a) ZOIs in shrub-free snowpacks and (b) IMP and BRAN layers identified in shrub snowpacks (see also Figure 6). Gray areas highlight the spectral range where calculated $k_e$ was fitted to measured $k_e$. Deviations at wavelengths >680 nm are interpreted as influence of buried branches.**

An important consequence of increased absorption by shrubs is a local heating effect. This local heating assumption was mentioned in Sturm et al. (2005) and Pomeroy et al. (2006) and is further supported by cursory observations on snow physical properties made during the field campaign in this study. During a warm spell on 19 and 20 November 2015, we observed that snow melt rates were increased in the direct vicinity of branches, forming a snowpack filled with holes (Fig. 8a and b). If shrub-induced radiative heating would have had a broader effect, the snowpack should melt more homogeneously than the observed swiss cheese snowpack. Localized melting around buried branches was also suggested by Sturm et al. (2005) and Pomeroy et al. (2006), which they considered to be an important factor for shrub spring-up in spring. In addition to the melt holes, we also found large clusters of melt-freeze grains attached to branches (Fig. 8c), indicating local melting.



When conditions were cold enough to prevent melting, the local radiative heating effect of branches resulted in the formation of pockets of depth hoar (or faceted crystals) around branches (Fig. 8d and e). Depth hoar are snow grains with a high

metamorphic degree (Akitaya, 1975) and are formed by high water vapor fluxes generated by strong temperature gradients in snowpacks. Strong vertical temperature gradients exist in the Arctic tundra in autumn, between the cold atmosphere and the relatively warmer soils. In the absence of shrubs, these temperature gradients typically form horizontal layers of depth hoar at the bottom of the Arctic snowpack. In the presence of shrubs, temperature gradients between the warmer branches and the colder snow nearby are increased, leading to enhanced depth hoar formation. As the effect of branches is very local,

however, this causes metamorphism only in the direct vicinity of branches, explaining the formation of depth hoar pockets rather than layers. This effect is particularly important for branches near the surface due to the proximity with the cold atmosphere and the higher irradiance. However, when depth hoar starts forming, its low thermal conductivity increases thermal gradients and further favours depth hoar formation so that the process may persist near branches even once they are deeply buried (this is also discussed in Domine et al., 2016).

The modifications of snow physical properties induced by buried branches are important because they influence the insulating effect of snow. In particular, depth hoar layers have very good insulating properties (Domine et al., 2016), while melt-freeze layers are poor insulators (Barrere et al., 2018). The insulating properties of a snowpack are critical for the survival of Arctic flora and fauna in winter (Berteaux et al., 2017; Domine et al., 2018), and directly impact the thermal regime of permafrost, which has important implications for ongoing climate change (Koven et al., 2013, Schuur et al., 2015).

Apart from these ecosystem-related consequences, shrub-induced modifications of snow physical properties are also disturbing the layered structure of the snowpack which is important for radiative transfer models calculating light propagation through snow under the presumption that snowpacks are plane-parallel media. It may be important to factor in these branch-induced structural disturbances in future studies simulating snow radiative transfer in mixed snowpacks.







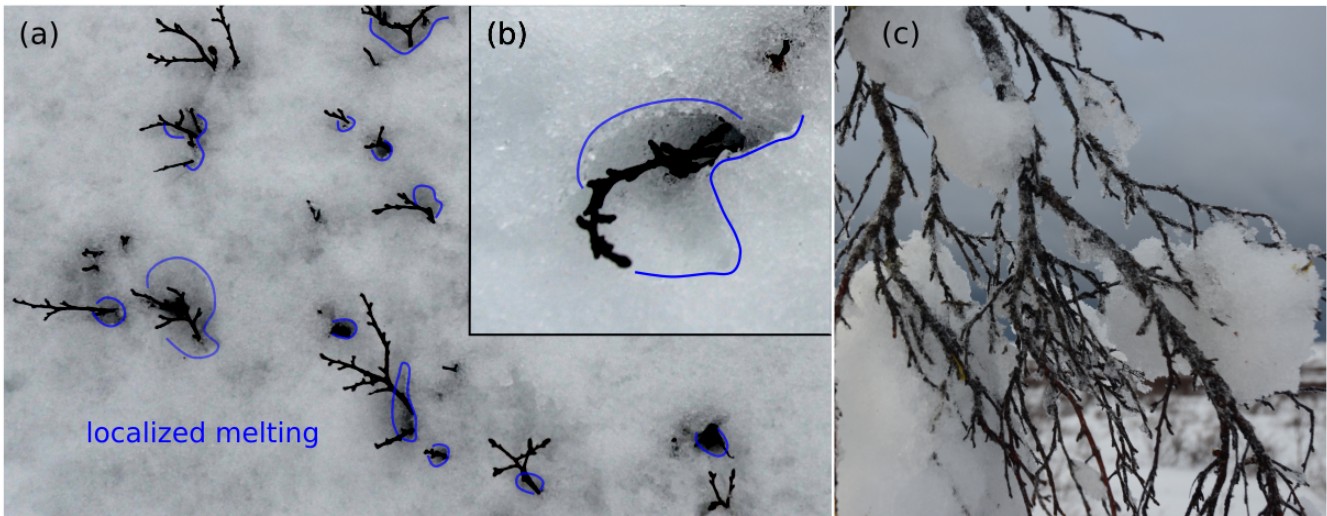

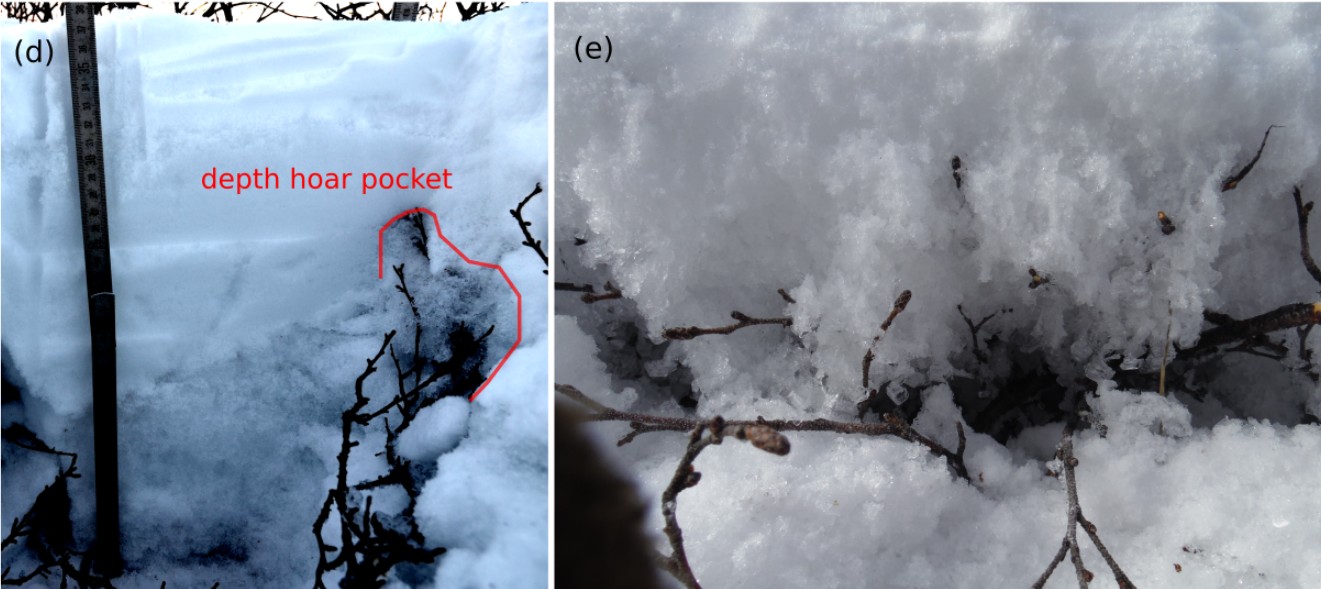

**Figure 8: Photos showing observations of localized snow melting around branches (a, b, c) and the formation of depth hoar pockets**
**around buried branches (d, e). Photos were taken during the measuring campaign from 29 Oct. to 6 Dec. 2015. In (d) the contrast**
**of the photo was increased to make the depth hoar pockets more visible.**

## 4.3 Source of high BC concentrations

The data presented here on shrub-free snowpacks were not intended to be an exhaustive study of impurities in snow in the

Umiujaq region, as their primary objective was to serve as a comparison to the measurements in snowpacks with shrubs. It is

nevertheless noteworthy that BC concentrations measured on 8 and 22 November were unexpectedly more than twice as

high as the median values reported for the rest of the Arctic, where concentrations outside Greenland lie around 20 ng g$^{-1}$,





with slightly higher values up to 60 ng g$^{-1}$ in Arctic Russia and Scandinavia (Doherty et al., 2010). High values similar to those measured here usually occur in mid-latitudes, for example in Northern China (117–1220 ng g$^{-1}$) (Wang et al., 2013) or the Chilean Andes (up to 100 ng g$^{-1}$) (Rowe et al., 2019), where the proximity to cities and industrial activities produce more

BC. The Arctic was usually found to be cleaner due to its distance to BC source regions (Skiles et al., 2016) and because BC concentrations have been continuously declining since industrial BC emissions in Europe and North America started decreasing in the early twentieth century (McConnell et al., 2007, Gong et al., 2010).

The high BC concentrations this study determined in Umiujaq cannot be due to forest fires because they do not happen in late fall or winter and are therefore most likely due to local anthropogenic sources early in the snow season. These could be

snowmobile traffic in the valley and, perhaps more importantly, the waste burning occurring ~2 km upwind from our study site (Fig. 2). Such local anthropogenic sources in the Arctic may become more influential as Arctic tourism keeps blooming, ship traffic keeps increasing, and northern communities keep growing. It is thus most likely that the contribution of BC emissions produced in the North, amongst others by waste burning, will increase in the near future and decrease snow albedo which could possibly advance the melt season by a few weeks (Tuzet et al., 2020). However, more research is necessary to

accurately quantify BC production in Umiujaq and in northern communities in general and to determine potential impacts. Nevertheless, our findings highlight that the anthropogenic footprint in the Arctic may be important and suggests that a cleaner waste management should be considered for the protection of northern communities and the ecosystem.

## 5 Conclusions

This study presented the first measurements of irradiance profiles in snowpacks with shrubs, together with complementary

irradiance measurements acquired in shrub-free snowpacks. Profiles measured in shrub-free snow were analysed to determine impurity type and impurity concentrations. For snow in Umiujaq, the main impurity, as inverted from a radiative transfer model, was black carbon (BC) which occurred in concentrations with large spatiotemporal variability. Some layers featured low concentrations (0–7 ng g$^{-1}$) while other layers had concentrations as high as 180–185 ng g$^{-1}$. High concentration layers were likely produced by the emission of nearby open-air waste burning. The high BC concentrations reported here

may be one of the first indicators that cleaner waste management plans are required to avoid the production of important BC concentrations from local sources in the Arctic. However, more research is required to draw firm conclusions over a longer period.

Irradiance profiles measured in snowpacks with shrubs showed that the impact of branches was weak and local. In some layers, light absorption depended primarily on BC concentrations and branches played only a minor role. In other layers,

coinciding with where branches were visible in snowpit photographs, the branch effect was more prominent, suggesting the local-effect hypothesis. This assumption was further supported by cursory observations of localized melting and depth hoar pockets forming in the vicinity of branches in the snowpack. The local modification of snow physical properties by branches


increases the heterogeneity of the snowpack and disturbs its plane-parallel structure. This heterogeneity should be considered by future research aiming to measure the radiative impact of shrubs *in situ* or simulate it with radiative transfer models.

**Authors contribution**

M. Belke-Brea designed the field experiments with contributions from all co-authors. F. Domine and G. Picard obtained funding. M. Belke-Brea and M. Barrere carried the experiments out. G. Picard developed the SnowMCML model code and performed the simulations. M. Belke-Brea prepared the manuscript with contributions from all co-authors.

**Competing interests**

The authors declare that they have no conflict of interests.

**Acknowledgements**

This work was funded by the BNP Paribas foundation (APT project), NSERC through the discovery grant program to FD and the French Polar Institute (IPEV) through program 1042 to FD. We thank the community of Umiujaq for their hospitality and support in the field. We are also grateful to the Centre d'Études Nordiques (CEN) for providing and maintaining the Umiujaq Research Station.

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



**Appendix A**

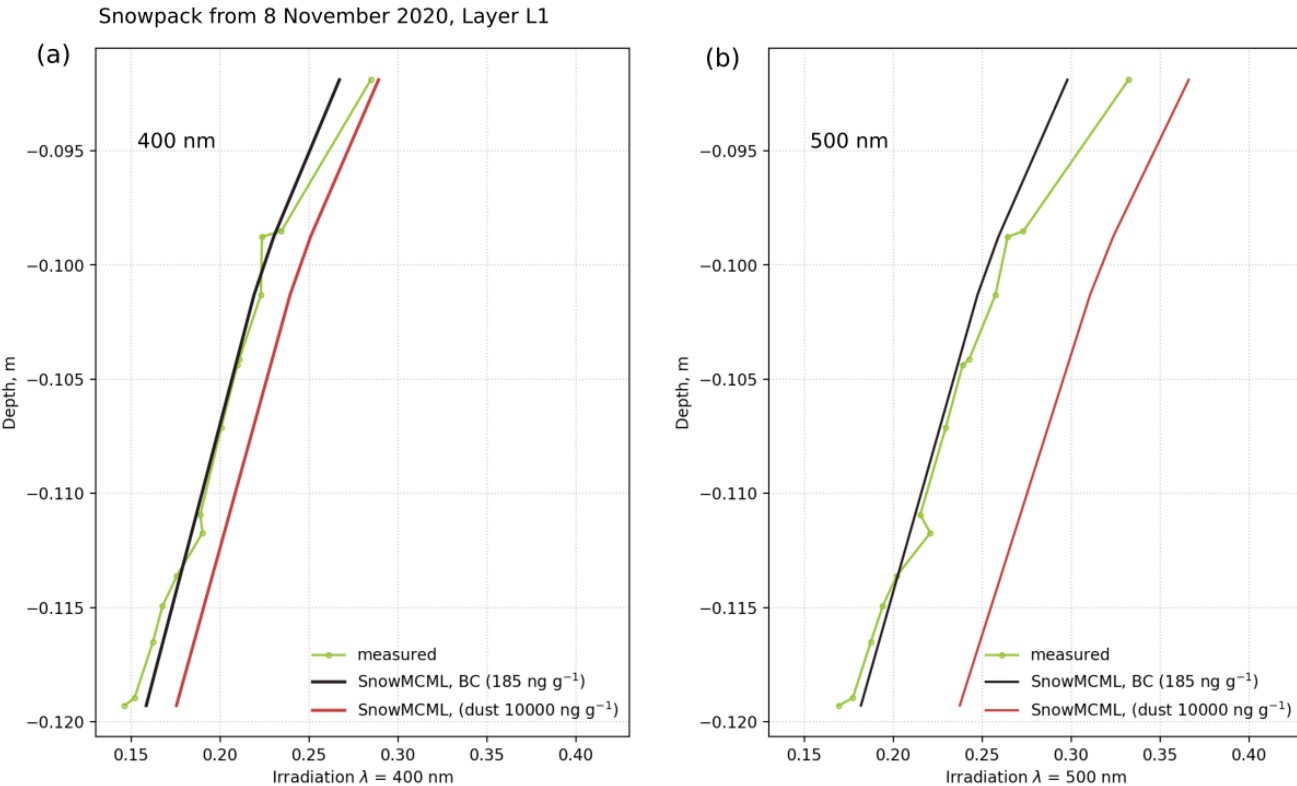

**Figure A1. SOLEXS irradiance profiles and SnowMCML simulations at 400 nm (a) and 500 nm (b) for 8 Nov. SnowMCML simulations were computed either with impurity type set to mineral dust (red plots) or BC (black plots). Using BC returns a good fit-quality independent of wavelength, while the fit with dust varies from 400 to 500 nm.**