# Peer review of "On the influence of erect shrubs on the irradiance profile in snow"

_Biogeosciences, 2020_

## Referee Comment (RC2)

**Review on**
**On the influence of erect shrubs on the irradiance profile in snow**

Maria Belke-Brea, Florent Domine, Ghislain Picard, Mathieu Barrere, and Laurent Arnaud

**Short summary**

The authors describe measurements and model results of snow properties and light extinction is snowpacks in Nunavik, Northern Quebec, Canada. The two main findings are (1) black carbon, and not mineral dust, dominates the light extinction within snowpacks without shrubs and (2) buried shrub branches influence radiation extinction in snow locally.

5 **General comments**

This study and its major findings are well supported by a high quality dataset. The findings are interesting and relevant although a quantification of the branch effect would have been desirable. Independent estimates of black carbon content from a laboratory analysis would also help in further studies. The paper is well written and includes all information needed to understand the results.

10     My major comment is, that the paper would greatly profit from a separation of the results and the discussion section (both new sections structured in the same subsections). I think that in this way, it would be easier for the reader to distinguish between new results and older references and general findings. The following paragraphs can be moved to the discussions almost as they are: lines 51–91, 367–372, 401–410, 469–472, 504–519, 525–527, 529–553, 567–587. Furthermore, I am missing some interesting discussion points that could be included in the new discussion section:

15 **Microtopography** Does it affect your results that your measurement sites were situated on a wind-exposed plateau (lines 114–115)? Snow properties are likely different between sheltered valley locations and plateaus, even without shrubs.

**Weather conditions** Did the weather conditions influence the results (lines 156–157)? Can the reduced quality of the profiles on 23 November be attributed to shadows on the snow?

**Spatial heterogeneity** I assume that even without shrubs, and definitely with shrubs, the snowpack is highly spatially hetero-
20     geneous. Did you do additional irradiance profiles not accompanied by snow pits (which are much more work)?

**ZOI3 and ZOI4** Please add some discussion on why ZOI3 and ZOI4 do not seem to support your main conclusion that black carbon is more important than mineral dust (Table 2).

**Non-local effects** Although radiative heating of branches buried in the snow have a mostly local effect (as you write), I think it would be good to discuss also possible non-local effects such as percolating melt water.

25 **Effect size** I understand, that it is difficult to quantify the effects of buried branches if the BC concentration is unknown. However, I would like to read some discussion on that topic. Maybe, you can also add a (very rough) estimate based on simplified assumptions? You write the effect is "weak" in abstract and conclusion, but it is not clear to me why.

As second general comment I would suggest to reduce the total number of figures while increasing the figure content as described in detail for each figure below. Figure quality could easily be improved with bigger fonts and joint axes for multiple

30 panels. Please use pdf as figure format (in every step of saving the figure) and not a pixel graphic (except for pictures) to avoid blurry text and allow the reader to zoom in.

**Specific comments**

**Short summary (online)** The short summary only includes the branches and not the black carbon results (which are also interesting).

35 **47–49** Please add another reference. Pelletier et al.,2018 do not discuss the light distribution in snow, rather snow depth in general and the formation of depth hoar.

**Introduction** The introduction is very informative but a bit too long. I suggest to move parts of lines 51–91 to the (new) Discussions section and remove detailed methods from lines 92–105.

**Figure 1** I find it hard to compare the two different datasets as the y-axis is very different and the grid does not match both

40 axis. Please compare reflectance of branches with reflectance of clean and dirty snow and show absorption of different particles per meter of snow in a second panel.

**115** Are all sites on the "wind-exposed plateau", even the sites with taller shrubs?

**Table 1** What do you mean by "average snow height"? As far as I understood, these are 7 snow pits with one snow height each. Did you measure snow height and shrub height at multiple points in the pit? If yes, please do not only show mean

45 values but also the variability. In the current table it is a bit confusing, why shrub height changes that much between the dates. Furthermore, I would like to see whether shrubs protruded the snow at all times and shrub sites or whether they were sometimes completely buried. Please also include the weather conditions during the radiation measurements. Please highlight the names of the layers analysed in your paper (like ZOI1, BRAN4) in this table.

**Figure 3** Which ZOI is this?

50 **Methods** It would be very helpful to see pictures of the measurement sites and landscape. Maybe this could be a second/third panel in Figure 2.

**Figure 4** What is AFEC? Please avoid new (any) abbreviations in figures. Figure 4 (a) is the same as Figure 7 (a). As Figure 7 is much more comprehensive, I suggest to add the additional line of Figure 4 (b) into Figure 7 (a) (in a different colour) and omit Figure 4. Also the other panels of Figure 7 could profit from an additional line showing simulations with mineral dust. Especially ZOI3 and ZOI4, which reveal similar/better results when including dust instead of BC. $R^2$ and RMSE can be omitted from the figure if you refer to Table 2.

**Table 2** Why are ZOI3 and ZOI4 separated by a line? In this table ZOI3 stands out as the fit with mineral dust is almost as good as the fit with black carbon; for ZOI4, only dust is even better that BC and the estimated BC concentration is very low. This was not mentioned in the text. It would be good to also show these examples in Figure 7. Please include this and the possible reasons in the discussion instead of just saying "Therefore, from now on we will assume that BC is the dominant impurity type for the remainder of this study." (lines 366–367)

**Figure 5** Please increase the font sizes. As all three panels have the same axes, it would be good to place them all in one row with joined y axis. this would save a lot of space and facilitate the comparison. I think it would be good to include results from the fit with dust in panel (c) as these ZOIs had a similar/better fit with dust than with BC. If the panel gets to busy with the additional information, you could add a fourth panel. I also suggest to combine this figure with Figure 6 in a similar way as Figure 7. In this way, it would be easier to compare extinction with and without branches.

**Figure 6** Please increase the font sizes. As all four panels have the same axes, it would be good to place them all in two rows with joined y axis or x axis, respectively (the labels IMP1, BRAN1,... can be moved into the plot area). In this way the size of the panels can be increased while the complete figure does not need more space.

**496–497** What about IMP1? It also seems to diverge.

**498** $R^2$ is always between 0 and 1, R ranges between -1 and 1. Did you confuse it with another variable? I assume that $R^2$ is Pearson's correlation coefficient (standard naming convention). This should be specified in the methods.

**Figure 7** The writing in this figure is too small and blurry. Please use joined the axes to allow bigger panels. Are the spectra averaged over the complete layers? Please also show ZOI4, IMP4, BRAN2 and BRAN3. I understood, that those were to noisy to perform calculations, but still I find it interesting to compare them visually to the other spectra. $R^2$ is always between 0 and 1, what are your numbers of IMP3 and BRAN1? Maybe remove the $R^2$ and RMSE from the figures and include them in Table 2.

**500–501** "..., calculated values can fit the observed values less well than a horizontal line (= the null hypothesis) which results in $R^2$ values below 0." This is a strange interpretation of the $R^2$. $R^2$ cannot be negative as it is squared (and you work with real numbers, I suppose). Negative values of R indicate that the dependent variable decreases if the independent variable increases and vice versa. A value of $R = -1$ is a very strong relationship and not a poor fit. I do not understand your comment about a horizontal line. This seems impossible as $k_e$ varies as a function of $\lambda$.

**454–455 & 488**  You write that IMP4 and BRAN3 had a worse quality ("log-irradiance profile was less regular", "signal-to-noise ratio was too low"). As shown in Figure 6c, IMP4 and BRAN3 were measured under sunny conditions. Would that be a possible explanation of the decreased quality of the profile measurements? I imagine that branches above the snow cast irregular shadows which influence the irradiance profile in different depth as compared to your reference sensor. By the way, was the reference sensor located above or below the branches? Please include the profiles in Figure 7, add information on the weather to Table 1, and discuss the effect of direct sunlight and shadows in a (new) discussion section.

**525–532**  You describe the heating effect as very local. However, I wonder what happens to the melt water. If the water percolates through the snow and refreezes in different parts, this would lead to a significant transfer of energy to deeper layers of the snowpack. Please discuss such non-local effects!

**529**  "broad" seems the wrong word. I think "non-local" would be more precise.

**Figure 8**  Panel (c) does not seem to fit the message of this figure. It looks like a branch on a tree above the snowpack. In this case, wind can also remove snow, not only localized melting. Maybe remove that panel (or explain it in more detail).

**567–587**  The (new) discussion section could start (rather than end) with this part, as results on BC are also shown before results on branches. It can also be merged with lines 401–410 and 367–372.

**594–596**  State briefly the possible implications of dirty Arctic snow.

**596–597**  Do not devalue your own study. You found important indications. Of course you can suggest further research, but rather in a positive phrase like: "Based on our results, we suggest further research on the regional and long-term importance of waste management in Arctic regions."

**598–604**  Some more implications of your results would be good here. Maybe you can mention (again) the snow insulating properties and their importance for permafrost/flora/fauna?

**598**  How come you classify the effect as "weak"? As far as I understood, you were not really able to quantify it. The estimated BC concentrations (especially at BRAN4) may be much higher that the "true" BC concentrations as, in the model, they include the branch effect at $400$–$450\,\mathrm{nm}$.

**605–608**  This does not seem to justify the co-authorship of F. Domine and L. Arnaud.

**References**  Please add the missing DOIs. The poor formatting of the references makes it hard to find details on papers.

**Figure A1**  Is Layer L1 the same as ZOI1? Please use consistent names and bigger fonts. The figure is a bit lost here in the appendix. How about combining it with Figure 3 (using the same example, of course)? Please change the word "plots" to "lines" ("red lines" and "black lines").

**General**  It would have been more convenient if you used hyperlinks so I could click on the references and links.

---

## Author Comment (AC1)

**Reply to reviewer RC1**

We thank the reviewer for very helpful and constructive comments. We have done our best to address the suggestions and will improve our manuscript accordingly. Our responses and the actions we will take are detailed below, embedded in the text of the reviews. Line numbers refer to the version that was submitted for review.

Belke-Brea et al. present interesting work on the difficult subjects of measuring and modelling light transmission in snow with buried shrubs. The introduction makes shrubs sound rather like invasive species that have only recently arrived in the tundra; they are actually natural (albeit expanding) components of tundra biomes. Trapping of snow by shrubs is likely to have more influence on the insulating properties of a snowpack, but absorption of light by buried branches clearly does have an influence and has received less attention. The title to could be modified to reflect that a lot of this paper is about the influence of soot.

Yes, a lot of the paper is about soot, but this really is not our focus. It would also be strange and probably confusing to mix shrubs and soot in the title, although we do recognize that part of the take-home message of the paper is about soot. At this point, we admit there must be some arbitrary character in choosing the title.

A wind rose would be a nice addition to Figure 2 in place of the wind direction arrows, if the AWS can provide.

Thank you for this good idea. The AWS provides both wind speed and direction and we will add a wind rose to Figure 2.

Do you have any information on the frequency and volume of waste burning in winter?

No, unfortunately not. The importance of the waste burning in winter emerged only when we started analyzing the data. Monitoring waste burning activities was thus not part of our measuring protocol in 2015. In any case, waste burning at Umiujaq is a random process without any precise organization.

The measured absorption coefficients in Figure 4 clearly cannot be fitted well by adding dust to the snow, but Figure 4(b) does not look like the best possible fit for 400-450 nm (a negative bias could be removed).

We verified our calculations and in Figure 4(b) the best-fit was indeed performed for the spectral range 350-450 nm instead of 400-450 nm. We will correct for that error and update Figure 4(b) with the corrected data.

How close together and how comparable were the sites for snow pits without shrubs?

The snow pits without shrubs were within a range of ~15 m from each other, and as the topography and wind exposure of all snow pit locations were similar, the snow pits were comparable. We will add this information in line 135.

Table 1 shows that snow depth increased by 7 cm between 22 and 28 November. Doesn't that mean that the clean snow in ZOI4 on 28 November was already on the ground when the snow was judged to be dirty on 22 November?

No, this is not necessarily the case because the Arctic snow cover is highly dynamic. Snow that accumulated on 22 November was probably re-distributed by strong winds, which are frequent in autumn, or melted partly or completely due to warm spells. It is thus very difficult to date the deposition of a given snow layer and the snow we measured on 22 and 28 November was most likely not the same. For more clarity we will add in the discussion section in line 354 that snow height at a given site may vary differently from snow height at a similar-looking site 15 m away, and that the snow layers measured on 22 and 28 November were most likely not the same layer.

Comparing with Figure 5, I think that the ZOI depths in Table 2 are wrong.

Thank you for highlighting this. Indeed, we made a mistake in Table 2 and will correct the ZOI depths there.

Above 700 nm, the increased grain size of depth hoar will have an effect of decreasing absorption coefficients. Could the large grains and voids in the snow around shrub branches act as pipes for transmission of near-infrared light (just to make modelling radiative transfer even harder)?

Light travelling in the air-filled voids around shrub branches would indeed be less absorbed compared to light travelling through snow. It is thus possible that more radiation reaches deeper snow layers than would be the case in a snowpack without voids. We would expect that this effect is very difficult to measure but it could be simulated with a full-3D Monte Carlo modelling approach.

---

## Author Comment (AC2)

**Reply to reviewer RC2**

We thank the reviewer for very helpful, detailed and constructive comments. We have done our best to address the suggestions and will improve our manuscript accordingly. Our responses and the actions we will take are detailed below, embedded in the text of the review. Line numbers refer to the version that was submitted for review.

**Short summary**

The authors describe measurements and model results of snow properties and light extinction is snowpacks in Nunavik, Northern Quebec, Canada. The two main findings are (1) black carbon, and not mineral dust, dominates the light extinction within snowpacks without shrubs and (2) buried shrub branches influence radiation extinction in snow locally.

**General comments**

This study and its major findings are well supported by a high quality dataset. The findings are interesting and relevant although a quantification of the branch effect would have been desirable. Independent estimates of black carbon content from a laboratory analysis would also help in further studies. The paper is well written and includes all information needed to understand the results.

We thank the reviewer for this encouraging comment. We agree that a quantification of the branch effect would have been desirable. Unfortunately, findings from field measurements can be unpredictable, especially for first-time measurements (like irradiation measurements in snowpacks with shrubs) and when acquisitions are taken under difficult meteorological conditions and in remote areas. We admit that there is an exploratory character to this investigation.

My major comment is, that the paper would greatly profit from a separation of the results and the discussion section (both new sections structured in the same subsections). I think that in this way, it would be easier for the reader to distinguish between new results and older references and general findings. The following paragraphs can be moved to the discussions almost as they are: lines 51–91, 367–372, 401–410, 469–472, 504–519, 525–527, 529–553, 567–587.

Separating results and discussion is in general preferable. However, in this case where results presentation is modified and optimized based on preliminary conclusions from previous results, separation would lead to repetitions. For example, the early conclusion that absorption by BC is important allows for a clearer and more focused presentation of subsequent results.

We will restructure our results and discussion sections to distinguish more clearly the results from the conclusions that were drawn from these results. We hope that like this it will become clearer what is result and what discussion.

Furthermore, I am missing some interesting discussion points that could be included in the new discussion section:

**Microtopography** Does it affect your results that your measurement sites were situated on a wind-exposed plateau (lines 114–115)? Snow properties are likely different between sheltered valley locations and plateaus, even without shrubs.

It is true, of course, that both vegetation and topography do affect accumulation and therefore snow properties. However, among the sites we studied there are no sheltered valleys and the microtopography is thus such that snow accumulation at the time of measurement was not all that variable. Since the variations are not very important within the relatively small area we studied, we cannot contribute any new conclusions based on the data set presented in this study. We will however add to our discussion in line 544 that microtopography was not variable and therefore did not have an important influence on variations in snow properties.

**Weather conditions** Did the weather conditions influence the results (lines 156–157)? Can the reduced quality of the profiles on 23 November be attributed to shadows on the snow?

Yes, shadows impact the quality of the measured irradiance profiles. Important are both, the shadow of the operator and the shadows created by protruding branches. If the operator stands opposite to the sun, he creates less shadow in the profile during sunny days compared to overcast days. This is counter-intuitive because shadows are not visible during overcast days. For branches that protrude close to the irradiance profile, they are likely to have an effect at depth. The depth of influence is ~ 1-2 x equal to the size of the shadows due to the diffusion.

The effect of shadows was not considered in the presented manuscript and we will add in lines 475 that sunny conditions and the effect of the shadow of branches have likely contributed to the high variability of the extinction coefficient.

**Spatial heterogeneity** I assume that even without shrubs, and definitely with shrubs, the snowpack is highly spatially heterogeneous. Did you do additional irradiance profiles not accompanied by snow pits (which are much more work)?

Measuring additional irradiance profiles which are not accompanied by snow pits would indeed allow to show the variability of irradiance on a larger area. However, without snowpits it would not be possible to interpret the measured variability. For this reason, such measurements were not done in our 2015 field campaign. Nevertheless, as the reviewer highlights in her comment, this exploratory study does suggest that it would be interesting to dig deeper into the variability of irradiance, and future studies should envisage this strategy.

**ZOI3 and ZOI4** Please add some discussion on why ZOI3 and ZOI4 do not seem to support your main conclusion that black carbon is more important than mineral dust (Table 2).

ZOI3 and ZOI4 are generally very clean snow layers with negligible amounts of light absorbing impurities. Given that the objective of the impurity analysis here was to determine the significant absorbers only, we excluded mineral dust from that list. In those snow layers where the spectral signature of dust returned a good model fit, mineral dust appeared only in trace amounts and had thus a very limited radiative effect. We therefore think that our conclusion holds true that BC is the most important absorber for the snowpack. We will add this explanation in line 353.

The reviewer is right, however, that for an exhaustive study on impurity type and concentration, it would be crucial to further analyse those trace amounts of dust. This, however, goes beyond the scope of this study which was aimed to better understand the effect of branches. An analysis on impurities would require different measurements as mentioned in line 371 ff.

**Non-local effects** Although radiative heating of branches buried in the snow have a mostly local effect (as you write), I think it would be good to discuss also possible non-local effects such as percolating melt water.

In the snowpits we observed indicators for melting and percolation, such as ice lenses and melt-freeze crystals, only close to branches. Percolation in snowpacks with shrubs is heterogeneous and would be initiated preferentially near branches. If branch-induced percolation is limited, it would thus be found mostly close to branches. Percolation and melt-freezing remain then local effects, which is what we observed in the snowpacks of this study. However, in a case of extensive melting, percolation may extend further from branches. We will add this discussion to the manuscript on line 544.

**Effect size** I understand, that it is difficult to quantify the effects of buried branches if the BC concentration is unknown. However, I would like to read some discussion on that topic. Maybe, you can also add a (very rough) estimate based on simplified assumptions? You write the effect is "weak" in abstract and conclusion, but it is not clear to me why.

From IMP1, IMP3, and BRAN1 in Figure 7 we can see that the presence of branches reduces light absorption for wavelengths > 680 nm compared to a snowpack with only snow and BC. This reduction of the absorption coefficient ranges at 700 nm between 2-12 $m^{-1}$ and at 800 nm between 12-27 $m^{-1}$. We will include this information in our discussion text in line 519. However, quantifying the effect of buried branches further will be difficult. We learned from the impurity analysis that BC concentrations can vary between 7 and 184 ng $g^{-1}$, and making a safe assumption on BC concentrations in our snow with branches is therefore impossible. We understand that this is frustrating, but the unexpected variation in BC is simply not allowing for a quantitative estimate of the branch effect.

Our mistake was to use "local" and "weak" interchangeably. There are layers in snowpacks with shrubs without a trace of a radiative impact of branches, like IMP2 in Figure 7, suggesting that the effect of branches in snow is not global but local. The reviewer is right, however, that the effect of branches may be strong at a local scale. We will remove the word "weak" from the manuscript and replace it with "local".

As second general comment I would suggest to reduce the total number of figures while increasing the figure content as described in detail for each figure below. Figure quality could easily be improved with bigger fonts and joint axes for multiple panels. Please use pdf as figure format (in every step of saving the figure) and not a pixel graphic (except for pictures) to avoid blurry text and allow the reader to zoom in.

As suggested by the reviewer, we will use joint axes for multiple panels and increase fonts were possible. For a potential publication, high quality vector graphics will be submitted solving the problem with blurriness. Our answers to comments below contain more details on figure modifications.

**Specific comments**

**Short summary (online)** The short summary only includes the branches and not the black carbon results (which are also interesting).

It is true that the short summary focuses on branches. This is because the aim and novelty of this study was to better understand the effect of branches, and given the limited room we decided to include only

the branch effect in the short summary. We will modify the last sentence of the short summary to state that BC was found to be the strongest absorber at our study sites.

**47–49** Please add another reference. Pelletier et al.,2018 do not discuss the light distribution in snow, rather snow depth in general and the formation of depth hoar.

We will add the publication of Dombrovsky et al., 2019 (DOI: 10.1016/j.jqsrt.2019.02.004 ).

**Introduction** The introduction is very informative but a bit too long. I suggest to move parts of lines 51–91 to the (new) Discussions section and remove detailed methods from lines 92–105.

Considering the interdisciplinary readership of Biogeosciences we decided to give precedence to a longer and more informative text in order to provide a suitable introduction to the physical, glaciological and biological aspects of this study. However, following the comment of the reviewer we will move the following lines of the introduction into a discussion section:
1) 62-68
2) 79-86

We feel that all other information in lines 51-91 is relevant to an introduction and to the full understanding of the methodology and results.

We will also remove the details on the method in lines 98-103.

**Figure 1** I find it hard to compare the two different datasets as the y-axis is very different and the grid does not match both axis. Please compare reflectance of branches with reflectance of clean and dirty snow and show absorption of different particles per meter of snow in a second panel.

The figure was designed to highlight the different absorption behaviour of shrub branches and impurities at wavelengths > 700 nm. However, we understand that the figure is difficult to read given the different y-axes. We suggest replacing the current figure with a figure showing co-albedo values of branches, snow with BC, snow with dust and clean snow.

We think showing co-albedo instead of albedo will allow for a more intuitive understanding of the reminder of the study, which focuses on absorption rather than reflectivity (although one is of course linked to the other ... ).

**115** Are all sites on the "wind-exposed plateau", even the sites with taller shrubs?

Yes, all sites are on the wind-exposed plateau. However, the North sides in Figure 2 tend to be a bit windier and to have slightly less snow. We will add this distinction in line 115 and add to Table 1 if the position of the snowpits was in the northern or southern part of the plateau.

**Table 1** What do you mean by "average snow height"? As far as I understood, these are 7 snow pits with one snow height each. Did you measure snow height and shrub height at multiple points in the pit? If yes, please do not only show mean values but also the variability.

The word "average" sneaked in here without a reason. As the reviewer understood correctly, each snow pit has one snow height. We apologize for that mistake and will remove "average" from our text.

In the current table it is a bit confusing, why shrub height changes that much between the dates.

Digging a snow pit and conducting the measurements within is a destructive process. Snow pits can therefore not be re-measured at the exact same location but have to be moved by a few meters each time. This is why the shrub height changes, as shrubs don't have the same height at each location. We will add this explanation in the new text.

Furthermore, I would like to see whether shrubs protruded the snow at all times and shrub sites or whether they were sometimes completely buried.

We are not sure to understand the reviewers comment here. We were not aiming at creating a time-series and therefore didn't take daily pictures of each snow pit which show whether branches were sometimes completely buried. The last column in Table 1 indicates the height of the protruding branches for those days when we measured the snowpits. We will rename that column to "Height of protruding branches" and add the information to the text by how much branches were protruding during the snowpit measurements.

Please also include the weather conditions during the radiation measurements. Please highlight the names of the layers analysed in your paper (like ZOI1, BRAN4) in this table.

As requested by the reviewer we will add 2 columns to Table 1, one each for the weather conditions and the names of the analyzed layers per snowpit.

**Figure 3** Which ZOI is this?

It is ZOI 2, from the profile measured on 22 November (the date is indicated at the top of the figure). We will add "Subsequently referred to as ZOI 2" to our caption.

**Methods** It would be very helpful to see pictures of the measurement sites and landscape. Maybe this could be a second/third panel in Figure 2.

As suggested, we will add a picture of the measurement sites in Figure 2 and submit a supplementary material document with more pictures of the landscape.

**Figure 4** What is AFEC? Please avoid new (any) abbreviations in figures.

The abbreviation AFEC will be removed and updated to $k_{e\_calc}$.

Figure 4 (a) is the same as Figure 7 (a). As Figure 7 is much more comprehensive, I suggest to add the additional line of Figure 4 (b) into Figure 7 (a) (in a different colour) and omit Figure 4.

True Figure 4(a) and Figure 7 (a) are the same. However, the message for each of the figures is very different and we would therefore keep both figures in the manuscript for more clarity.

Also the other panels of Figure 7 could profit from an additional line showing simulations with mineral dust. Especially ZOI3 and ZOI4, which reveal similar/better results when including dust instead of BC. R 2 and RMSE can be omitted from the figure if you refer to Table 2.

Please see our answer two comments further down.

**Table 2** Why are ZOI3 and ZOI4 separated by a line?

The separation between ZOI3 and ZOI4 by a line is a graphical error, we apologize for not having noticed it and will correct this.

In this table ZOI3 stands out as the fit with mineral dust is almost as good as the fit with black carbon; for ZOI4, only dust is even better that BC and the estimated BC concentration is very low. This was not mentioned in the text. It would be good to also show these examples in Figure 7. Please include this and the possible reasons in the discussion instead of just saying "Therefore, from now on we will assume that BC is the dominant impurity type for the remainder of this study." (lines 366–367)

The text fails to mention that in ZOI4 the "dust only" simulation returns a slightly better fit. We will add this to the text.

However, we feel that our conclusion is still true that "BC is the only significant absorber in snow without shrubs and that absorption due to dust is negligible. (355-356)" As we mention in lines 369ff, this doesn't mean that dust is not at all present in the snow, but that its effect is too weak to consider in this analysis which focuses on the radiative branch effect. We think that considering dust, and including the dust plots of ZOI3 and 4 to Figure 7, would only complexify the problem. Adding a second impurity type with unknown concentration to the branch analysis will not help our understanding of the radiative effect of branches. We will therefore supply the graphs with dust simulations in a supplementary material section. We will also stress in line 366 that BC is the principal absorber for the snowpack in general, as opposed to the principal absorber in any given layer.

**Figure 5** Please increase the font sizes. As all three panels have the same axes, it would be good to place them all in one row with joined y axis. this would save a lot of space and facilitate the comparison. I think it would be good to include results from the fit with dust in panel (c) as these ZOIs had a similar/better fit with dust than with BC. If the panel gets to busy with the additional information, you could add a fourth panel.

As suggested by the reviewer, we will place the panels in a row and use a joint y-axis. For the reasons given above, we are hesitant to include the results of the dust simulations as well, as this study really focuses on the effect of branches and adding dust would complexify the problem without adding new insights to the branch effect analysis.

However, we understand that for many readers it is interesting to see the possible effects of dust and we will add the requested additional information for dust for panel (c) in the supplementary material.

I also suggest to combine this figure with Figure 6 in a similar way as Figure 7. In this way, it would be easier to compare extinction with and without branches.

We understand how merging Figure 5 and 6 makes it easier to compare simulation results for snowpits with shrubs and shrub-free snowpits. However, in this case we feel that merging both figures is complicated because it creates a very loaded figure. As both figures have different messages, we think that keeping them separated makes it easier to understand the presented study.

**Figure 6** Please increase the font sizes. As all four panels have the same axes, it would be good to place them all in two rows with joined y axis or x axis, respectively (the labels IMP1, BRAN1,... can be moved into the plot area). In this way the size of the panels can be increased while the complete figure does not need more space.

We will modify Figure 6 as suggested by the reviewer.

**496–497** What about IMP1? It also seems to diverge.

We are not sure to fully understand the reviewers comment. We agree that for IMP1 the measured and calculated curves diverge, this was written in lines 494-496. We will modify the text in line 504 so that it clearly states how we explain the divergence in IMP1 as well as in IMP3, BRAN1 and BRAN4.

**498** R 2 is always between 0 and 1, R ranges between -1 and 1. Did you confuse it with another variable? I assume that R 2 is Pearson's correlation coefficient (standard naming convention). This should be specified in the methods.

As the reviewer indicates, in standard naming convention R is the Pearson's correlation coefficient and $R^2$ is the coefficient of determination. In some cases $R^2$ can indeed be equal to the square of R, however it is actually calculated by:

$$R^2 = 1 - (SS_{total} / SS_{res}), \qquad \text{Eq. (1)}$$

where $SS_{total}$ is the sum of squares of the vertical distance of all points to a horizontal line (called the null hypothesis) and $SS_{res}$ is the sum of squares of the vertical distance of all points to the model (see also sketch below). The horizontal line for the null hypothesis corresponds to the mean of the Y values. Eq. (1) shows that $R^2$ is not actually the square of anything, although the name intuitively suggests it.

[Figure]

It is true that $R^2$ usually varies between 0 and 1, because the variation explained by a model should be better than that explained by the simple mean of the Y values, and thus $R^2$ should be greater than 0 (otherwise you could just use the mean and have a better performing model). However, $R^2$ can be negative when a constraint is applied to a model. For example, when forcing an intercept or, as was done in the presented study, when fitting the model to only part of the spectral data. Applying such a constraint can result in a model which performs worse than the null hypothesis and which therefore has a negative $R^2$ value. In the presented study, the spectral curves for snowpacks with branches (e.g. Figure 7, BRAN1) had an almost horizontal form and it was therefore particularly easy to obtain a negative $R^2$, even for models that visually have a moderately good fit for most of the spectral range (e.g. Figure 7, IMP1).

What negative $R^2$ values indicate is that the chosen model is no good in reproducing the measured data. However, the model we used in the presented study is missing one component, i.e. the absorbing effect of branches, and it is therefore not surprising that it has a poor fit. The misfit between model and measured data was intentionally used to highlight the effect of branches.

We will revise the text to clarify the meaning of a negative $R^2$ value without going into too much detail. Otherwise, a very good explanation on the $R^2$ value can also be found in the cited publication of Motulsky and Christopoulos, 2003.

**Figure 7** The writing in this figure is too small and blurry. Please use joined the axes to allow bigger panels.

We apologize if the figure annotations were difficult to read. We will submit vector graphics for a potential final publication which will reduce the blurriness. As suggested we will join the y-axes for each of the rows in the figure to make the plots bigger.

Are the spectra averaged over the complete layers?

To obtain the spectral extinction coefficients ($k_{e\_meas}$) as shown in Figure 7 we used the method developed by Tuzet et al. 2019, which is also illustrated in Figure 3 and explained in lines 274 to 281. The spectra are not averaged over the layers but correspond to the slope of the linear regression made for the irradiance data in the layers. The slope obtained from the linear regression corresponds to the extinction coefficient $k_{e\_meas}$.

Please also show ZOI4, IMP4, BRAN2 and BRAN3. I understood, that those were to noisy to perform calculations, but still I find it interesting to compare them visually to the other spectra.

The spectral extinction coefficient as shown in Figure 7 can only be obtained in layers where irradiance decreases linearly and where the extinction coefficient is a constant (Figure 3). As the profiles of ZOI4, IMP4, BRAN2 and BRAN3 are too noisy, such layers with a linearly decreasing irradiance don't exist and it is unfortunately impossible to calculate the spectral extinction curves for those layers. Note please that irradiance at ZOI4 has a linear decrease at 400 nm (see Figure 5c), but becomes noisy at longer wavelengths for which snow is more absorbent and which therefore penetrates less deep.

R 2 is always between 0 and 1, what are your numbers of IMP3 and BRAN1?

Please see the comment above.

Maybe remove the R 2 and RMSE from the figures and include them in Table 2.

We think that having $R^2$ and RMSE values in the same figure is more convenient because it avoids that the reader has to scroll back and forth between the figure and the table.

**500–501** "..., calculated values can fit the observed values less well than a horizontal line (= the null hypothesis) which results in R 2 values below 0." This is a strange interpretation of the R 2 . R 2 cannot be negative as it is squared (and you work with real numbers, I suppose). Negative values of R indicate that the dependent variable decreases if the independent variable increases and vice versa. A value of R = −1 is a very strong relationship and not a poor fit. I do not understand your comment about a horizontal line. This seems impossible as k e varies as a function of λ.

We hope that with the explanation given above this is clearer now. As mentioned above we will revise the text to clarify the meaning of a negative $R^2$ value. We will also include the equation to calculate $R^2$ as shown in Eq. (1) in this document.

**454–455 & 488** You write that IMP4 and BRAN3 had a worse quality ("log-irradiance profile was less regular", "signal-to-noise ratio was too low"). As shown in Figure 6c, IMP4 and BRAN3 were measured under sunny conditions. Would that be a possible explanation of the decreased quality of the profile measurements? I imagine that branches above the snow cast irregular shadows which influence the irradiance profile in different depth as compared to your reference sensor.

The different influence of shadows on measured profiles vs. the reference sensor would not change the quality of the measurement. The reference sensor was only used to ascertain that the illumination conditions were stable during the measuring period. We recorded only the % change in the incoming light intensity. These reference measurements were not used to correct the measured irradiance profiles in any way, but only to discard profiles where the light intensity varied more than 3% during acquisition.

Nevertheless, during sunny conditions the shadows cast by branches on the snow surface will indeed have an impact on the quality of the irradiance profile with depth. In general, the effect of shadows is attenuated with depth, while the area affected by the shadow increases. For example, a point shadow at the surface has a cone-shaped effect with depth, i.e. a circle that is extending with depth (with radius = depth). Within the snowpack, shadows cast by individual branches at the snow surface create a complex 3D field of light because the effect of the different shadows overlap. Thus for a given point (x, y), the *I(z)* profile decreases and increases because of variations in the influence of different shadows and open areas. As the reviewer suggests, this could be one explanation for the irregular profile and variations observed in the extinction coefficient in IMP4 and BRAN3 in Figure 6(c), and we will include this in the discussion text.

We will also add in line 475 that *I(z)* can be influenced by 1) snow absorption and its impurities, 2) branches buried in the snow near the profile due to the extra absorption by the branch, and 3) in sunny conditions, the complex 3D field of shadows cast by protruding branches.

By the way, was the reference sensor located above or below the branches? Please include the profiles in Figure 7, add information on the weather to Table 1, and discuss the effect of direct sunlight and shadows in a (new) discussion section.

The reference sensor was located on a branch-free snow surface next to our measuring spots. However, this has no impact on our measurements because, as mentioned above, the sensor was only used to monitor the change in intensity of incoming radiation.

The information on weather will be added to Table 1.

**525–532** You describe the heating effect as very local. However, I wonder what happens to the melt water. If the water percolates through the snow and refreezes in different parts, this would lead to a significant transfer of energy to deeper layers of the snowpack. Please discuss such non-local effects!

The transfer of latent heat by melted, percolated and re-frozen water is a considerable factor in the snow energy budget. For example, the amount of transferred latent energy by only 1 g of water is 1 to 2 orders of magnitude larger than the transfer of sensible heat through a snowpack with a thermal gradient of 20K m$^{-1}$ and a thermal conductivity of 0.05 W m$^{-1}$ K$^{-1}$. In our manuscript we have not made any distinction between the sensible and latent heat transfer processes associated with buried branches and we will add a discussion in line 544.

The latent-heat effect caused by branches may be non-local if melting expands spatially and affects the whole snowpack. However, this is not what we observed in late fall (November and early December), when melt-freeze indicators were found close to branches. We think intuitively that when melting occurs only because of radiative heating of branches, then meltwater production would be local and limited. This is especially true in late fall, when irradiance is limited, too. It may not be true in spring (April and May), when irradiance increases. Based on the observations in this study, we have to conclude that both the latent and sensible heat effects of branches are local in late fall. However, future studies conducted in spring may find a non-local effect, when the radiative heating of branches increases with increasing incoming radiation.

**529** "broad" seems the wrong word. I think "non-local" would be more precise.

As suggested "broad" will be replaced by "non-local".

**Figure 8** Panel (c) does not seem to fit the message of this figure. It looks like a branch on a tree above the snowpack. In this case, wind can also remove snow, not only localized melting. Maybe remove that panel (or explain it in more detail).

Panel (c) actually shows a branch of a shrub which was buried by snow and which we cut-off before taking the photo. Attached to the branches are large clusters of melt-freeze grains which melted locally and refrozen around the branch. For a better understanding, we will present a zoom-out version of the photo showing the entire branch as well as a zoom-in on the melt-freeze grains. We will also explain in the caption that this is a picture of a buried branch extracted from the snowpack for the purpose of taking a picture.

**567–587** The (new) discussion section could start (rather than end) with this part, as results on BC are also shown before results on branches. It can also be merged with lines 401–410 and 367–372.

This is one interesting possibility for organizing the manuscript. However, we wish to keep the focus on our initial goal: investigate the effect of branches of light propagation in snow. Since the effect of impurities was unexpected but nevertheless had to be discussed, we prefer to discuss that point at the end, not the beginning, to stay consistent with our main objectives.

**594–596** State briefly the possible implications of dirty Arctic snow.

As suggested we will state that impurities in Arctic snow accelerate snow melting in spring and can also amplify the impact of warm spells in autumn.

**596–597** Do not devalue your own study. You found important indications. Of course you can suggest further research, but rather in a positive phrase like: "Based on our results, we suggest further research on the regional and long-term importance of waste management in Arctic regions."

Thank you for this comment. We will change the sentence as suggested.

**598–604** Some more implications of your results would be good here. Maybe you can mention (again) the snow insulating properties and their importance for permafrost/flora/fauna?

As suggested we will add that the local modifications of snow physical properties and thus of snow microstructure may impact insulating properties and thus affect the thermal regime of permafrost as well as flora and fauna.

**598** How come you classify the effect as "weak"? As far as I understood, you were not really able to quantify it. The estimated BC concentrations (especially at BRAN4) may be much higher that the "true" BC concentrations as, in the model, they include the branch effect at 400–450 nm.

As mentioned above our mistake was to use "local" and "weak" interchangeably and we will remove the word "weak" from the manuscript and replace it with "local".

**605–608** This does not seem to justify the co-authorship of F. Domine and L. Arnaud.

L. Arnaud and G. Picard developed the instrument used in this study. F. Domine obtained funding, which was critical for the execution of this project, and was also actively involved in designing the field protocol and writing this manuscript. We will modify the author's contribution text to contain a more detailed explanation of each author's contribution.

**References** Please add the missing DOIs. The poor formatting of the references makes it hard to find details on papers.

Thank you for highlighting this, the missing DOIs will be added.

**Figure A1** Is Layer L1 the same as ZOI1? Please use consistent names and bigger fonts. The figure is a bit lost here in the appendix. How about combining it with Figure 3 (using the same example, of course)? Please change the word "plots" to "lines" ("red lines" and "black lines").

We will change the figure as suggested. We will also group this figure with the other dust-related figures in the supplementary material.

**General** It would have been more convenient if you used hyperlinks so I could click on the references and links.

---

## Author Response (AR1)

**Responses to reviewers**

We thank the reviewers for very helpful, detailed and constructive comments. We have done our best to address their suggestions and to improve our manuscript accordingly. Our responses to each comment are detailed below in green, embedded in the text of their reviews. Revised text is in *blue italics*, with line numbers referring to the tracked changes version. We hope these responses are clear, and we look forward to submitting the revised manuscript.

Reviewer #1

Belke-Brea et al. present interesting work on the difficult subjects of measuring and modelling light transmission in snow with buried shrubs. The introduction makes shrubs sound rather like invasive species that have only recently arrived in the tundra; they are actually natural (albeit expanding) components of tundra biomes. Trapping of snow by shrubs is likely to have more influence on the insulating properties of a snowpack, but absorption of light by buried branches clearly does have an influence and has received less attention. The title to could be modified to reflect that a lot of this paper is about the influence of soot.

Yes, a lot of the paper is about soot, but this really is not our focus. It would also be strange and probably confusing to mix shrubs and soot in the title, although we do recognize that part of the take-home message of the paper is about soot. At this point, we admit there must be some arbitrary character in choosing the title.

A wind rose would be a nice addition to Figure 2 in place of the wind direction arrows, if the AWS can provide.

Thank you for this good idea. The AWS provides both wind speed and direction.

We added a wind rose to Figure 2, using wind speed and direction data from October to September measured in the time period 2012-2019.

Do you have any information on the frequency and volume of waste burning in winter?

No, unfortunately not. The importance of the waste burning in winter emerged only when we started analyzing the data. Monitoring waste burning activities was thus not part of our measuring protocol in 2015. In any case, waste burning at Umiujaq is a random process without any precise organization.

The measured absorption coefficients in Figure 4 clearly cannot be fitted well by adding dust to the snow, but Figure 4(b) does not look like the best possible fit for 400-450 nm (a negative bias could be removed).

We verified our calculations and in Figure 4(b) the best-fit was indeed performed for the spectral range 350-450 nm instead of 400-450 nm. We corrected for that error and update Figure 4(b) as well as the dust simulation results in Table 2.

How close together and how comparable were the sites for snow pits without shrubs?

The snow pits without shrubs were within a range of ~10 m from each other, and as the topography and wind exposure of all snow pit locations were similar, the snow pits were comparable. Text added in line 141 -142:

*"All shrub-free sites were within a range of ~10 m from each other and are considered comparable with similar topography and wind exposure."*

Table 1 shows that snow depth increased by 7 cm between 22 and 28 November. Doesn't that mean that the clean snow in ZOI4 on 28 November was already on the ground when the snow was judged to be dirty on 22 November?

No, this is not necessarily the case because the Arctic snow cover is highly dynamic. Snow that accumulated on 22 November was probably re-distributed by strong winds, which are frequent in autumn, or melted partly or completely due to warm spells. It is thus very difficult to date the deposition of a given snow layer and the snow we measured on 22 and 28 November was most likely not the same. For more clarity we will added text in line 422–426:

*"It is important to note that despite the small increase in snow height from 22 and 28 November (only 7 cm), it is unlikely that the same snow measured on 22 November was still the surface snow layer on 28 November because of snow re-distribution by strong winds and melt events during warm spells. This explains how a snowpack with high BC could be measured on 22 November, while one week later on 28 November the snowpack was found to be clean with low BC concentrations."*

Comparing with Figure 5, I think that the ZOI depths in Table 2 are wrong.

Thank you for highlighting this. Indeed, we made a mistake in Table 2.
All depth indication in Table 2 were corrected.

Above 700 nm, the increased grain size of depth hoar will have an effect of decreasing absorption coefficients. Could the large grains and voids in the snow around shrub branches act as pipes for transmission of near-infrared light (just to make modelling radiative transfer even harder)?

Light travelling in the air-filled voids around shrub branches would indeed be less absorbed compared to light travelling through snow. It is thus possible that more radiation reaches deeper snow layers than would be the case in a snowpack without voids. We would expect that this effect is very difficult to measure but it could be simulated with a full-3D Monte Carlo modelling approach.

Reviewer #2

**Short summary**

The authors describe measurements and model results of snow properties and light extinction is snowpacks in Nunavik, Northern Quebec, Canada. The two main findings are (1) black carbon, and not mineral dust, dominates the light extinction within snowpacks without shrubs and (2) buried shrub branches influence radiation extinction in snow locally.

**General comments**

This study and its major findings are well supported by a high quality dataset. The findings are interesting and relevant although a quantification of the branch effect would have been desirable. Independent estimates of black carbon content from a laboratory analysis would also help in further studies. The paper is well written and includes all information needed to understand the results.

We thank the reviewer for this encouraging comment. We agree that a quantification of the branch effect would have been desirable. Unfortunately, findings from field measurements can be unpredictable, especially for first-time measurements (like irradiation measurements in snowpacks with shrubs) and when acquisitions are taken under difficult meteorological conditions and in remote areas. We admit that there is an exploratory character to this investigation.

My major comment is, that the paper would greatly profit from a separation of the results and the discussion section (both new sections structured in the same subsections). I think that in this way, it would be easier for the reader to distinguish between new results and older references and general findings. The following paragraphs can be moved to the discussions almost as they are: lines 51–91, 367–372, 401–410, 469–472, 504–519, 525–527, 529–553, 567–587.

Separating results and discussion is in general preferable. However, in this case where results presentation is modified and optimized based on preliminary conclusions from previous results, separation would lead to repetitions. For example, the early conclusion that absorption by BC is important allows for a clearer and more focused presentation of subsequent results.

We restructured our results and discussion sections to distinguish more clearly the results from the conclusions that were drawn from these results. In particular, we joined the sections on impurity types and concentrations to group all results from the $k_e$ and SnowMCML analysis in lines 344–390. The discussion of these results was then grouped in lines 401–428. Similarly, we joined the description of the optical appearance of the IMP and BRAN layer (444–497), and the discussion of these results (lines 507–548). We hope that like this it becomes clearer what is result and what discussion.

Furthermore, I am missing some interesting discussion points that could be included in the new discussion section:

**Microtopography** Does it affect your results that your measurement sites were situated on a wind-exposed plateau (lines 114–115)? Snow properties are likely different between sheltered valley locations and plateaus, even without shrubs.

It is true, of course, that both vegetation and topography do affect accumulation and therefore snow properties. However, among the sites we studied there are no sheltered valleys and the microtopography is thus such that snow accumulation at the time of measurement was not all that variable. Since the variations are not very important within the relatively small area we studied, we cannot contribute any new conclusions based on the data set presented in this study. We added the following text to lines 575–578:

*"The effect of branches on melting and depth hoar formation may vary in different topographical settings because microtopography impacts total snow accumulation but also snow density, with wind exposed areas having denser snowpacks than wind-sheltered areas. However, topography was similar in all our sites so no topography-induced impact could be studied."*

**Weather conditions** Did the weather conditions influence the results (lines 156–157)? Can the reduced quality of the profiles on 23 November be attributed to shadows on the snow?

Yes, shadows impact the quality of the measured irradiance profiles. Important are both, the shadow of the operator and the shadows created by protruding branches. If the operator stands opposite to the sun, he creates less shadow in the profile during sunny days compared to overcast days. This is counter-intuitive because shadows are not visible during overcast days. For branches that protrude close to the irradiance profile, they are likely to have an effect at depth. The depth of influence is ~ 1-2 x equal to the size of the shadows due to the diffusion.

The effect of shadows was not considered in the presented manuscript. We added text in lines 518 – 526:

*"The irregularities may be caused through direct light absorption by branches, but also by branch shadows cast at the surface or within the snowpack. Especially during sunny conditions, the shadows cast by branches at the snow surface can have an impact on the quality of the irradiance profile with depth. The effect of shadows is attenuated with depth, while the area affected by the shadow increases. For example, a point shadow at the surface has a cone-shaped effect with depth, i.e. a circle that is extending with depth (with radius = depth). As a consequence, shadows cast by individual branches at the snow surface create a complex 3D field of light within the snowpack because the effect of the different shadows overlap. Thus, for a given point (x, y), the I(z) profile decreases and increases because of variations in the influence of different shadows and open areas,*

*creating irregular profiles. Therefore, branch shadows could be one explanation for the irregular profile and variations observed in the extinction coefficient, particularly in BRAN3, which was a sunny day."*

**Spatial heterogeneity** I assume that even without shrubs, and definitely with shrubs, the snowpack is highly spatially heterogeneous. Did you do additional irradiance profiles not accompanied by snow pits (which are much more work)?

Measuring additional irradiance profiles which are not accompanied by snow pits would indeed allow to show the variability of irradiance on a larger area. However, without snowpits it would not be possible to interpret the measured variability. For this reason, such measurements were not done in our 2015 field campaign. Nevertheless, as the reviewer highlights in her comment, this exploratory study does suggest that it would be interesting to dig deeper into the variability of irradiance, and future studies should envisage this strategy.

**ZOI3 and ZOI4** Please add some discussion on why ZOI3 and ZOI4 do not seem to support your main conclusion that black carbon is more important than mineral dust (Table 2).

ZOI3 and ZOI4 are generally very clean snow layers with negligible amounts of light absorbing impurities. Given that the objective of the impurity analysis here was to determine the significant absorbers only, we excluded mineral dust from that list. In those snow layers where the spectral signature of dust returned a good model fit, mineral dust appeared only in trace amounts and had thus a very limited radiative effect. We therefore think that our conclusion holds true that BC is the most important absorber for the snowpack.

The reviewer is right, however, that for an exhaustive study on impurity type and concentration, it would be crucial to further analyse those trace amounts of dust. This, however, goes beyond the scope of this study which was aimed to better understand the effect of branches.

We added text in the results and discussion paragraphs. New text in the results paragraph was added in lines 364 – 367:

*"For the low impurity zones (ZOI3 and 4), using BC only, dust only or dust and BC in Eq. (3) returned consistently low impurity concentrations with similarly good fits for all impurity types. For example, in ZOI3 the best fit ($R^2 = 0.98$) was achieved by using BC only, while using dust only reduced the fit slightly ($R^2 = 0.96$). For ZOI4 the best fit ($R^2 = 0.92$) was achieved by using dust only, while using BC only reduced the fit slightly ($R^2 = 0.91$)."*

New text in the discussion paragraph was added in lines 401 – 412:

*"We conclude from the $k_e$ analysis and SnowMCML simulations that BC is the significant absorber in snow without shrubs. However, for the snowpack measured on 28 November, using dust instead of BC in simulations also returned a good fit and it is likely that some trace amounts of dust, coming from the cuestas surrounding the Tasiapik valley, were also present in the snow. Nevertheless, the overall impurity concentrations on 28 November were very low and the good fit for the SnowMCML profile simulated without impurities suggests that impurity impact on absorption was negligible in those cases. This is, of course, a simplification, and a chemical analysis of LAPs in snow would nicely complement these optical studies, but this is left for future work. Based on our current data, we conclude that light extinction in snow without shrubs is best modelled with BC as the main impurity type. This is also consistent with other studies in the Arctic who have found BC to be the main impurity type in Arctic snow (e.g. Doherty et al., 2010; Wang et al., 2013). Moreover, the open-air waste burning near our study area was probably an important additional BC source (Fig. 2). For the purpose of this study we will therefore assume that BC is the dominant impurity type in the Umiujaq snowpack."*

**Non-local effects** Although radiative heating of branches buried in the snow have a mostly local effect (as you write), I think it would be good to discuss also possible non-local effects such as percolating melt water.

In the snowpits we observed indicators for melting and percolation, such as ice lenses and melt-freeze crystals, only close to branches. Percolation in snowpacks with shrubs is heterogeneous and would be initiated preferentially near branches. If branch-induced percolation is limited, it would thus be found mostly close to branches. Percolation and melt-freezing remain then local effects, which is what we observed in the snowpacks of this study. However, in a case of extensive melting, percolation may extend further from branches. We added this in lines 562-563:

*"A non-local effect could be caused during warm spells in autumn and in spring, when more extensive melting may be possible, leading to percolation that would affect the whole snowpack, but this was not observed."*

**Effect size** I understand, that it is difficult to quantify the effects of buried branches if the BC concentration is unknown. However, I would like to read some discussion on that topic. Maybe, you can also add a (very rough) estimate based on simplified assumptions? You write the effect is "weak" in abstract and conclusion, but it is not clear to me why.

From IMP1, IMP3, and BRAN1 in Figure 7 we can see that the presence of branches reduces light absorption for wavelengths > 680 nm compared to a snowpack with only snow and BC. This reduction of the absorption coefficient ranges at 700 nm between 2-12 $m^{-1}$ and at 800 nm between 12-27 $m^{-1}$. However, quantifying the effect of buried branches further will be difficult. We learned from the impurity analysis that BC concentrations can vary between 8 and 185 ng $g^{-1}$, and making a safe assumption on BC concentrations in our snow with branches is therefore impossible. We understand that this is frustrating, but the unexpected variation in BC is simply not allowing for a quantitative estimate of the branch effect.

New text on branch-induced light absorption changes at wavelengths > 680 nm was added in lines 546 – 548:

*"Moreover, the magnitude of the radiative effect of branches varies also as a function of wavelength, as indicated by the results from the $k_e$ analysis where branches, compared to BC, reduce the absorption coefficient at 700 nm by 2 to 12 $m^{-1}$ and at 800 nm by 12 to 27 $m^{-1}$."*

Our mistake was to use "local" and "weak" interchangeably. There are layers in snowpacks with shrubs without a trace of a radiative impact of branches, like IMP2 in Figure 7, suggesting that the effect of branches in snow is not global but local. The reviewer is right, however, that the effect of branches may be strong at a local scale. We removed 'weak' in line 19 and 623.

As second general comment I would suggest to reduce the total number of figures while increasing the figure content as described in detail for each figure below. Figure quality could easily be improved with bigger fonts and joint axes for multiple panels. Please use pdf as figure format (in every step of saving the figure) and not a pixel graphic (except for pictures) to avoid blurry text and allow the reader to zoom in.

As suggested by the reviewer, we used joint axes for multiple panels, increased fonts and included vector graphics were possible. For a potential publication, high quality graphics will be submitted as individual files solving the problem with blurriness. Our answers to comments below contain more details on figure modifications.

**Specific comments**

**Short summary (online)** The short summary only includes the branches and not the black carbon results (which are also interesting).

It is true that the short summary focuses on branches. This is because the aim and novelty of this study was to better understand the effect of branches, and given the limited room we decided to include only the branch effect in the short summary. We modified the short summary and it states now that BC was found to be the strongest absorber at our study sites.

**47–49** Please add another reference. Pelletier et al.,2018 do not discuss the light distribution in snow, rather snow depth in general and the formation of depth hoar.

The publication of Dombrovsky et al., 2019 (DOI: 10.1016/j.jqsrt.2019.02.004 ) was added in line 48.

**Introduction** The introduction is very informative but a bit too long. I suggest to move parts of lines 51–91 to the (new) Discussions section and remove detailed methods from lines 92–105.

Considering the interdisciplinary readership of Biogeosciences we decided to give precedence to a longer and more informative text in order to provide a suitable introduction to the physical, glaciological and biological aspects of this study. However, following the comment of the reviewer we removed text from lines 62-68 and from lines 94-98.

**Figure 1** I find it hard to compare the two different datasets as the y-axis is very different and the grid does not match both axis. Please compare reflectance of branches with reflectance of clean and dirty snow and show absorption of different particles per meter of snow in a second panel.

The figure was designed to highlight the different absorption behaviour of shrub branches and impurities at wavelengths > 700 nm. However, we understand that the figure is difficult to read given the different y-axes. We changed the Figure 1 to show albedo of dirty and clean snow as well as branch albedo.

**115** Are all sites on the "wind-exposed plateau", even the sites with taller shrubs?

Yes, all sites are on the wind-exposed plateau. However, the North sides in Figure 2 tend to be a bit windier and to have slightly less snow. We added this distinction in lines 110-112:

*"All measuring sites were situated on a wind-exposed plateau, with sites situated in the northern part of the plateau (Fig. 2 and Table 1) being slightly windier and with slightly less snow."*

We also added to Table 1 if the position of the snowpits was in the northern or southern part of the plateau.

**Table 1** What do you mean by "average snow height"? As far as I understood, these are 7 snow pits with one snow height each. Did you measure snow height and shrub height at multiple points in the pit? If yes, please do not only show mean values but also the variability.

The word "average" sneaked in here without a reason. As the reviewer understood correctly, each snow pit has one snow height. We apologize for that mistake. We removed "average" from Table 1.

In the current table it is a bit confusing, why shrub height changes that much between the dates.

Digging a snow pit and conducting the measurements within is a destructive process. Snow pits can therefore not be re-measured at the exact same location but have to be moved by a few meters each time. This is why the shrub height changes, as shrubs don't have the same height at each location. We add an explanatory sentence in lines 136-140:

*"Snow and irradiance measurements are destructive and could therefore only be taken once for each site. Measurements in snowpacks with shrubs were conducted on 3, 9, 14 and 23 November. Snow height at these sites varied between 43 and 63 cm, shrub height varied between 60 and 100 cm and the height of protruding branches varied between 10 and 42 cm (Table 1). The differences in shrub height (Table 1) indicate the intra-site variability."*

Furthermore, I would like to see whether shrubs protruded the snow at all times and shrub sites or whether they were sometimes completely buried.

We are not sure to understand the reviewers comment here. We were not aiming at creating a time-series and therefore didn't take daily pictures of each snow pit which show whether branches were sometimes completely buried. The last column in Table 1 indicates the height of the protruding branches for those days when we measured the snowpits. We renamed the column to "Height of protruding branches" and added information on how much branches were protruding during the snowpit measurements in lines 138-139:

*"Snow height at these sites varied between 43 and 63 cm, shrub height varied between 60 and 100 cm and the height of protruding branches varied between 10 and 42 cm (Table 1)."*

Please also include the weather conditions during the radiation measurements. Please highlight the names of the layers analysed in your paper (like ZOI1, BRAN4) in this table.

As requested by the reviewer we added 2 columns to Table 1, one each for the weather conditions and the names of the analyzed layers per snowpit.

**Figure 3** Which ZOI is this?

It is ZOI 2, from the profile measured on 22 November (the date is indicated at the top of the figure). We added 'ZOI 2' to Figure 3a, and included the following sentence in the caption in line 326:

*"The blue shaded area is subsequently referred to as ZOI 2."*

**Methods** It would be very helpful to see pictures of the measurement sites and landscape. Maybe this could be a second/third panel in Figure 2.

As suggested, we added two pictures of the measurement sites in Figure 2 and submitted a supplementary material document with more pictures of the landscape in Figure A1.

**Figure 4** What is AFEC? Please avoid new (any) abbreviations in figures.

The abbreviation AFEC was be removed in Figure 4 and updated to $k_{e\_calc}$.

Figure 4 (a) is the same as Figure 7 (a). As Figure 7 is much more comprehensive, I suggest to add the additional line of Figure 4 (b) into Figure 7 (a) (in a different colour) and omit Figure 4.

True Figure 4 (a) and Figure 7 (a) are the same. However, the message for each of the figures is very different and we would therefore keep both figures in the manuscript for more clarity.

Also the other panels of Figure 7 could profit from an additional line showing simulations with mineral dust. Especially ZOI3 and ZOI4, which reveal similar/better results when including dust instead of BC. R 2 and RMSE can be omitted from the figure if you refer to Table 2.

Please see our answer two comments further down.

**Table 2** Why are ZOI3 and ZOI4 separated by a line?

The separation between ZOI3 and ZOI4 by a line is a graphical error, we apologize for not having noticed it. The line has been removed in Table 2.

In this table ZOI3 stands out as the fit with mineral dust is almost as good as the fit with black carbon; for ZOI4, only dust is even better that BC and the estimated BC concentration is very low. This was not mentioned in the text. It would be good to also show these examples in Figure 7. Please include this and the possible reasons in the discussion instead of just saying "Therefore, from now on we will assume that BC is the dominant impurity type for the remainder of this study." (lines 366–367)

The text fails to mention that in ZOI4 the "dust only" simulation returns a slightly better fit. We added the following in lines 365-367:

*"For example, in ZOI3 the best fit ($R^2$ = 0.98) was achieved by using BC only, while using dust only reduced the fit slightly ($R^2$ = 0.96). For ZOI4 the best fit ($R^2$ = 0.92) was achieved by using dust only, while using BC only reduced the fit slightly ($R^2$ = 0.91)."*

However, we feel that our conclusion is still true that BC is the only significant absorber in snow without shrubs and that absorption due to dust is negligible. This doesn't mean that dust is not at all present in the snow, but that its effect is too weak to consider in this analysis which focuses on the radiative branch effect. We think that considering dust, and including the dust plots of ZOI3 and 4 to Figure 7, would only complexify the problem. Adding a second impurity type with unknown concentration to the branch analysis will not help our understanding of the radiative effect of branches. We supplied the graphs with dust simulations in a supplementary material document in Figure A2.1 and Figure A2.2. We also added text in lines 401–412 that stresses that BC is the principal absorber for the snowpack in general, as opposed to the principal absorber in any given layer:

*"We conclude from the $k_e$ analysis and SnowMCML simulations that BC is the significant absorber in snow without shrubs. However, for the snowpack measured on 28 November, using dust instead of BC in simulations also returned a good fit and it is likely that some trace amounts of dust, coming from the cuestas surrounding the Tasiapik valley, were also present in the snow. Nevertheless, the overall impurity concentrations on 28 November were very low and the good fit for the SnowMCML profile simulated without impurities suggests that impurity impact on absorption was negligible in those cases. This is, of course, a simplification, and a chemical analysis of LAPs in snow would nicely complement these optical studies, but this is left for future work. Based on our current data, we conclude that light extinction in snow without shrubs is best modelled with BC as the main impurity type. This is also consistent with other studies in the Arctic who have found BC to be the main impurity type in Arctic snow (e.g. Doherty et al., 2010; Wang et al., 2013). Moreover, the open-air waste burning near our study area was probably an important additional BC source (Fig. 2). For the purpose of this study we will therefore assume that BC is the dominant impurity type in the Umiujaq snowpack."*

**Figure 5** Please increase the font sizes. As all three panels have the same axes, it would be good to place them all in one row with joined y axis. this would save a lot of space and facilitate the comparison. I think it would be good to include results from the fit with dust in panel (c) as these ZOIs had a similar/better fit with dust than with BC. If the panel gets to busy with the additional information, you could add a fourth panel.

As suggested by the reviewer, we placed the panels in a row and used joint y-axis. For the reasons given above, we are hesitant to include the results of the dust simulations as well, as this study really focuses on the effect of branches and adding dust would complexify the problem without adding new insights to the branch effect analysis.

However, we understand that for many readers it is interesting to see the possible effects of dust and we added the requested additional information for dust in the supplementary material in Figure A2.2.

I also suggest to combine this figure with Figure 6 in a similar way as Figure 7. In this way, it would be easier to compare extinction with and without branches.

We understand how merging Figure 5 and 6 makes it easier to compare simulation results for snowpits with shrubs and shrub-free snowpits. However, in this case we feel that merging both figures is complicated because it creates a very loaded figure. As both figures have different messages, we think that keeping them separated makes it easier to understand the presented study.

**Figure 6** Please increase the font sizes. As all four panels have the same axes, it would be good to place them all in two rows with joined y axis or x axis, respectively (the labels IMP1, BRAN1,... can be moved into the plot area). In this way the size of the panels can be increased while the complete figure does not need more space.

Figure 6 was modified as suggested by the reviewer.

**496–497** What about IMP1? It also seems to diverge.

We are not sure to fully understand the reviewer's comment. We agree that for IMP1 the measured and calculated curves diverge, this was written in lines 484–485. We modified the text in lines 529–533 to clearly state how the divergence in IMP1, IMP3, BRAN1 and BRAN4 are explained:

*"In contrast, for IMP1, IMP3, BRAN1 and BRAN 4 the $k_{e\_meas}(\lambda)$ curves displayed a drop at wavelengths > 680 nm which was not visible in the $k_{e\_calc}(\lambda)$ curves. We interpret this drop as a strong indicator of the influence of branches (Fig. 7), because reflectivity measurements for Arctic shrub branches showed that branches are highly absorbing at 400–500 nm, but that reflectivity increases slightly at 500 nm and then even more sharply at 680 nm (Juszak et al. 2014) (Fig. 1)."*

**498** R 2 is always between 0 and 1, R ranges between -1 and 1. Did you confuse it with another variable? I assume that R 2 is Pearson's correlation coefficient (standard naming convention). This should be specified in the methods.

As the reviewer indicates, in standard naming convention R is the Pearson's correlation coefficient and $R^2$ is the coefficient of determination. In some cases, $R^2$ can indeed be equal to the square of R, however it is actually calculated by:

$$R^2 = 1 - (SS_{total} / SS_{res}),$$ Eq. (1)

where $SS_{total}$ is the sum of squares of the vertical distance of all points to a horizontal line (called the null hypothesis) and $SS_{res}$ is the sum of squares of the vertical distance of all points to the model (see also sketch below). The horizontal line for the null hypothesis corresponds to the mean of the Y values. Eq. (1) shows that $R^2$ is not actually the square of anything, although the name intuitively suggests it.

[Figure]

It is true that $R^2$ usually varies between 0 and 1, because the variation explained by a model should be better than that explained by the simple mean of the Y values, and thus $R^2$ should be greater than 0 (otherwise you could just use the mean and have a better performing model). However, $R^2$ can be negative when a constraint is applied to a model. For example, when forcing an intercept or, as was done in the presented study, when fitting the model to only part of the spectral data. Applying such a constraint can result in a model which performs worse than the null hypothesis and which therefore has a negative $R^2$ value. In the presented study, the spectral curves for snowpacks with branches (e.g. Figure 7, BRAN1) had an almost horizontal form and it was therefore particularly easy to obtain a negative $R^2$, even for models that visually have a moderately good fit for most of the spectral range (e.g. Figure 7, IMP1).

What negative $R^2$ values indicate is that the chosen model is no good in reproducing the measured data. However, the model we used in the presented study is missing one component, i.e. the absorbing effect of branches, and it is therefore not surprising that it has a poor fit. The misfit between model and measured data was intentionally used to highlight the effect of branches.

We revised the text to clarify the meaning of a negative $R^2$ value without going into too much detail. Otherwise, a very good explanation on the $R^2$ value can also be found in the cited publication of Motulsky and Christopoulos, 2003. The revised text and the equation to calculate $R^2$ are in lines 498–506:

*"Note the $R^2$ can be negative as it is not actually a square but is calculated with:*

$$R^2 \ = \ 1 - (SS_{total} \ / \ SS_{res}), \hspace{4cm} (4)$$

*where SStotal is the sum of squares of the vertical distance of all points to a horizontal line (called the null hypothesis) and SSres is the sum of squares of the vertical distance of all points to the model (Motulsky and Christopoulos, 2003). Negative $R^2$ indicate a poorly performing model (Motulsky and Christopoulos, 2003), which can happen when constraints are applied to a model. Here, we constrained the fit of $k_{e\_meas}(\lambda)$ and $k_{e\_calc}(\lambda)$ to the range 400–450 nm while performing model evaluation for a much larger range. The model used in the presented study is missing one component, i.e. the absorbing effect of branches, and it is thus not surprising that it has a poor fit. The misfit between model and measured data was intentionally used to study the effect of branches."*

**Figure 7** The writing in this figure is too small and blurry. Please use joined the axes to allow bigger panels.

We apologize if the figure annotations were difficult to read. We will submit vector graphics for a potential final publication which will reduce the blurriness. As suggested we will join the y-axes for each of the rows in the figure to make the plots bigger.

Are the spectra averaged over the complete layers?

To obtain the spectral extinction coefficients ($k_{e\_meas}$) as shown in Figure 7 we used the method developed by Tuzet et al. 2019, which is also illustrated in Figure 3 and explained in lines 283 to 286. The spectra are not averaged over the layers but correspond to the slope of the linear regression made for the irradiance data in the layers. The slope obtained from the linear regression corresponds to the extinction coefficient $k_{e\_meas}$.

Please also show ZOI4, IMP4, BRAN2 and BRAN3. I understood, that those were to noisy to perform calculations, but still I find it interesting to compare them visually to the other spectra.

The spectral extinction coefficient as shown in Figure 7 can only be obtained in layers where irradiance decreases linearly and where the extinction coefficient is a constant (Figure 3). As the profiles of ZOI4, IMP4, BRAN2 and BRAN3 are too noisy, such layers with a linearly decreasing irradiance don't exist and it is unfortunately impossible to calculate the spectral extinction curves for those layers. Note please that irradiance at ZOI4 has a linear decrease at 400 nm (see Figure 5c), but becomes noisy at longer wavelengths for which snow is more absorbent and which therefore penetrates less deep.

R 2 is always between 0 and 1, what are your numbers of IMP3 and BRAN1?

Please see the comment above.

Maybe remove the R 2 and RMSE from the figures and include them in Table 2.

We think that having $R^2$ and RMSE values in the same figure is more convenient because it avoids that the reader has to scroll back and forth between the figure and the table.

**500–501** "..., calculated values can fit the observed values less well than a horizontal line (= the null hypothesis) which results in R 2 values below 0." This is a strange interpretation of the R 2 . R 2 cannot be negative as it is squared (and you work with real numbers, I suppose). Negative values of R indicate that the dependent variable decreases if the independent variable increases and vice versa. A value of R = −1 is a very strong relationship and not a poor fit. I do not understand your comment about a horizontal line. This seems impossible as k e varies as a function of λ.

We hope that with the explanation given above this is clearer now. As mentioned above we revised the text in lines 489–497 to clarify the meaning of a negative $R^2$ value, where we also included the equation to calculate $R^2$ as shown in Eq. (1) in this document.

**454–455 & 488** You write that IMP4 and BRAN3 had a worse quality ("log-irradiance profile was less regular", "signal-to-noise ratio was too low"). As shown in Figure 6c, IMP4 and BRAN3 were measured under sunny conditions. Would that be a possible explanation of the decreased quality of the profile measurements? I imagine that branches above the snow cast irregular shadows which influence the irradiance profile in different depth as compared to your reference sensor.

The different influence of shadows on measured profiles vs. the reference sensor would not change the quality of the measurement. The reference sensor was only used to ascertain that the illumination conditions were stable during the measuring period. We recorded only the % change in the incoming light intensity. These reference measurements were not used to correct the measured irradiance profiles in any way, but only to discard profiles where the light intensity varied more than 3% during acquisition.

Nevertheless, during sunny conditions the shadows cast by branches on the snow surface will indeed have an impact on the quality of the irradiance profile with depth. We added the following text in lines 518–526:

*"The irregularities may be caused through direct light absorption by branches, but also by branch shadows cast at the surface or within the snowpack. Especially during sunny conditions, the shadows cast by branches at the snow surface can have an impact on the quality of the irradiance profile with depth. The effect of shadows is attenuated with depth, while the area affected by the shadow increases. For example, a point shadow at the surface has a cone-shaped effect with depth, i.e. a circle that is extending with depth (with radius = depth). As a consequence, shadows cast by individual branches at the snow surface create a complex 3D field of light within the snowpack because the effect of the different shadows overlap. Thus, for a given point (x, y), the I(z) profile decreases and increases because of variations in the influence of different shadows and open areas, creating irregular profiles. Therefore, branch shadows could be one explanation for the irregular profile and variations observed in the extinction coefficient, particularly in BRAN3, which was a sunny day."*

By the way, was the reference sensor located above or below the branches? Please include the profiles in Figure 7, add information on the weather to Table 1, and discuss the effect of direct sunlight and shadows in a (new) discussion section.

The reference sensor was located on a branch-free snow surface next to our measuring spots. However, this has no impact on our measurements because, as mentioned above, the sensor was only used to monitor the change in intensity of incoming radiation.

The information on weather was added to Table 1.

525–532 You describe the heating effect as very local. However, I wonder what happens to the melt water. If the water percolates through the snow and refreezes in different parts, this would lead to a significant transfer of energy to deeper layers of the snowpack. Please discuss such non-local effects!

The transfer of latent heat by melted, percolated and re-frozen water is a considerable factor in the snow energy budget. For example, the amount of transferred latent energy by only 1 g of water is 1 to 2 orders of magnitude larger than the transfer of sensible heat through a snowpack with a thermal gradient of 20K $m^{-1}$ and a thermal conductivity of 0.05 W $m^{-1}$ $K^{-1}$.

The latent-heat effect caused by branches may be non-local if melting expands spatially and affects the whole snowpack. However, this is not what we observed in late fall (November and early December), when melt-freeze indicators were found close to branches. We think intuitively that when melting occurs only because of radiative heating of branches, then meltwater production would be local and limited. This is especially true in late fall, when irradiance is limited, too. It may not be true in spring (April and May), when irradiance increases. Based on the observations in this study, we have to conclude that both the latent and sensible heat effects of branches are local in late fall. However, future studies conducted in spring may find a non-local effect, when the radiative heating of branches increases with increasing incoming radiation. We added the potential for a non-local effect in lines 562–563:

*"A non-local effect could be caused during warm spells in autumn and in spring, when more extensive melting may be possible, leading to percolation that would affect the whole snowpack, but this was not observed."*

529 "broad" seems the wrong word. I think "non-local" would be more precise.

As suggested "broad" was replaced by "non-local" in line 558.

Figure 8 Panel (c) does not seem to fit the message of this figure. It looks like a branch on a tree above the snowpack. In this case, wind can also remove snow, not only localized melting. Maybe remove that panel (or explain it in more detail).

Panel (c) actually shows a branch of a shrub which was buried by snow and which we cut-off before taking the photo. Attached to the branches are large clusters of melt-freeze grains which melted locally and refrozen around the branch. For a better understanding, we present in Figure 8c a zoom-out version of the photo showing the entire branch. Figure 8d shows a zoom-in on the melt-freeze grains. We also added in the caption that Figure 8c shows a picture of a buried branch extracted from the snowpack for the purpose of taking a picture.

**567–587** The (new) discussion section could start (rather than end) with this part, as results on BC are also shown before results on branches. It can also be merged with lines 401–410 and 367–372.

This is one interesting possibility for organizing the manuscript. However, we wish to keep the focus on our initial goal: investigate the effect of branches of light propagation in snow. Since the effect of impurities was unexpected but nevertheless had to be discussed, we prefer to discuss that point at the end, not the beginning, to stay consistent with our main objectives.

**594–596** State briefly the possible implications of dirty Arctic snow.

As suggested we stated in lines 637–638 that impurities in Arctic snow accelerate snow melting in spring and can also amplify the impact of warm spells in autumn:

*"Impurities in Arctic snow accelerate snow melting in spring and can also amplify the impact of warm spells in autumn."*

**596–597** Do not devalue your own study. You found important indications. Of course you can suggest further research, but rather in a positive phrase like: "Based on our results, we suggest further research on the regional and long-term importance of waste management in Arctic regions."

Thank you for this comment. We changed the sentence as suggested in lines 638–640.

**598–604** Some more implications of your results would be good here. Maybe you can mention (again) the snow insulating properties and their importance for permafrost/flora/fauna?

As suggested we added the following sentence to the text in lines 626–628:

*"The local modification of snow physical properties by branches and the resulting changes in snow microstructure impact the insulating properties of snow with potential consequences for Arctic flora and fauna as well as the thermal regime of permafrost."*

**598** How come you classify the effect as "weak"? As far as I understood, you were not really able to quantify it. The estimated BC concentrations (especially at BRAN4) may be

much higher that the "true" BC concentrations as, in the model, they include the branch effect at 400–450 nm.

As mentioned above our mistake was to use "local" and "weak" interchangeably and we will remove the word "weak" from the manuscript and replace it with "local".

**605–608** This does not seem to justify the co-authorship of F. Domine and L. Arnaud.

L. Arnaud and G. Picard developed the instrument used in this study. F. Domine obtained funding, which was critical for the execution of this project, and was also actively involved in designing the field protocol and writing this manuscript. The authors contribution section in lines 643–646 has been changed to the following:

*"FD and GP designed the research project. MBB designed the field experiments with contributions from all co-authors. FD and GP obtained funding. GP developed the Snow MCML model code and performed the simulations. LA and GP developed the SOLEX instrument used in this study. MBB and MB carried the experiments out. MBB wrote the paper with inputs from FD and comments from all co-authors."*

**References** Please add the missing DOIs. The poor formatting of the references makes it hard to find details on papers.

Thank you for highlighting this, the missing DOIs were added in the references.

**Figure A1** Is Layer L1 the same as ZOI1? Please use consistent names and bigger fonts. The figure is a bit lost here in the appendix. How about combining it with Figure 3 (using the same example, of course)? Please change the word "plots" to "lines" ("red lines" and "black lines").

The figure was replaced by Figure A2.2 in the supplementary material showing now dust simulations for all profiles in the shrub-free snowpacks.

**General** It would have been more convenient if you used hyperlinks so I could click on the references and links.

We changed the DOIs in the references to hyperlinks to make access to papers easier.

---

## Author Response (AR2)

**Suggestions for revision or reasons for rejection (will be published if the paper is accepted for final publication)**

We thank the reviewer for her thorough reading and all her suggestions and help! Please find below detailed answers on how we addressed the suggested corrections. For better readability we show the reviewer's comments in blue and our answers in green.

Thanks to the authors for answering to each comment in detail and for improving the manuscript. I have a few more minor points which would be good to address before final publishing.

As I suggested in my first review, a separation of a results and a discussion section would be very helpful. In the new version, the structure of the results improved and it is more clear now which parts are results and which are discussion. However, sections 4.1 and 4.2 are very long and need a subdivision with headings for improved readability.

The sections were subdivided with the following headings:
4.1 Impurity type and concentration in snow without shrubs
    4.1.1 Results from the ke analysis
    4.1.2 Results from SnowMCML simulations
    4.1.3 BC as significant absorber

4.2 Insights into the radiative effect of buried branches
    4.2.1 Comparison of log-irradiance profiles with SnowMCML simulations
    4.2.2 Spectral shape of $k_{e\_meas}(\lambda)$ vs. $k_{e\_calc}(\lambda)$
    4.2.3 Radiative effect of buried branches
    4.2.4 Local heating effect and impact on snow physical properties

The quality of the figures should be improved further:
(I) same font sizes in all figures (please consider the final figure width as specified in the journals guidelines); the fonts are still too small in Figures 3, 4, 5(only layer labels), 6(only layer labels), and 7 while they are too big in Figure 1 and 5(some of the labels); in Figure 6 the label 'Depth, m' has different font sizes in both rows;

Font sizes where changed as suggested. However, as we don't know the final typesetting of the manuscript it is possible that further adaption will be necessary during the author proof readings.

in Figure 8, the labels (like (a) and (e)) are in different fonts
Labels were set to different colors as the backgrounds are different. We set all fonts to same color and highlighted the text with white squares instead.

(II) remove the thin grey lines in Figures 2, A2.1 and A2.2,
We are not sure which grey lines the reviewer is referring to. Is it the grid lines in the plots and map?

(III) Figure 4: include meaning of shaded area and star in legend or caption,
The meaning of the shaded area and star were included in the figure.

(IV) Figure 7: remove x-axis labels and x-tick labels from the top row and shift it down to the other two rows.
We maintained the x-axis label here because the two rows belong to different sections ((a) and (b)) of the figure. The section annotations were missing in the last manuscript which was an oversight on our part – we apologize for that mistake.

Consider changing from supplement to appendix. Appendices are part of the manuscript whereas supplements are published along with the manuscript. Since you are referring to the figures in the main text, I suggest to use the appendix format, since it makes it easier to access the extra figures.

We agree with the reviewer but due to high costs of publication we decided to keep the additional material as supplements.

Move equation (4) (lines 488-494) to the methods part.
We moved the Equation (4) to lines 295 – 298.